# ROBUST MULTIVARIATE TIME-SERIES FORECASTING: ADVERSARIAL ATTACKS AND DEFENSE MECHANISMS

**Linbo Liu[1] , Youngsuk Park[1], Trong Nghia Hoang[2]∗, Hilaf Hasson[1], Jun Huan[1]**
[1]AWS AI Labs, [2]Washington State University,
{linbol, pyoungsu, hashilaf, lukehuan}@amazon.com, trongnghia.hoang@wsu.edu

## ABSTRACT

This work studies the threats of adversarial attack on multivariate probabilistic forecasting models and viable defense mechanisms. Our studies discover a new attack pattern that negatively impact the forecasting of a target time series via making strategic, sparse (imperceptible) modifications to the past observations of a small number of other time series. To mitigate the impact of such attack, we have developed two defense strategies. First, we extend a previously developed randomized smoothing technique in classification to multivariate forecasting scenarios. Second, we develop an adversarial training algorithm that learns to create adversarial examples and at the same time optimizes the forecasting model to improve its robustness against such adversarial simulation. Extensive experiments on real-world datasets confirm that our attack schemes are powerful and our defense algorithms are more effective compared with baseline defense mechanisms.

## 1 INTRODUCTION

Understanding the robustness for time-series models has been a long-standing issue with applications across many disciplines such as climate change (Mudelsee, 2019), financial market analysis (Andersen et al., 2005; Hallac et al., 2017), down-stream decision systems in retail (Böse et al., 2017), resource planning for cloud computing (Park et al., 2019; 2020), and optimal control of vehicles (Kim et al., 2020). In particular, the notion of robustness defines how sensitive the model output is when authentic data is (potentially) perturbed with noises. In practice, as observation data are often corrupted by measurement noises, it is important to develop statistical forecasting models that are less sensitive to such noises (Brown, 1957; Brockwell & Davis, 2009; Taylor & Letham, 2018) or more stable against outliers that might arise from such corruption (Connor et al., 1994; Gelper et al., 2010; Liu & Zhang, 2021; Wang & Tsay, 2021). However, these approaches have not considered the possibility of adversarial noises which are strategically created to mislead the model rather than being sampled from a known distribution.

As a matter of fact, vulnerabilities against such adversarial noises have been previously pointed out (Szegedy et al., 2013; Goodfellow et al., 2014b) in classification. In practice, it has been shown that human-imperceptible adversarial perturbation can alter classification outcomes of a deep learning (DL) model, revealing a severe threat to many safety-critical systems . As such a risk is associated with the high capacity to fit complex data pattern of DL, we postulate that similar threats might also occur in forecasting where modern DL-based forecasting models (Rangapuram et al., 2018; Salinas et al., 2020; Lim et al., 2020; Wang et al., 2019; Park et al., 2022) have become the dominant approach. For example, to mislead the forecasting of a particular stock, the adversaries might attempt to alter some features external to the stock's financial valuation to maximize the gap between predictions of its values on authentic and altered features. The feasibility of such an adversarial attack has been recently demonstrated with tweet messages (Xie et al., 2022) on a text-based stock forecasting.

Motivated by these real scenarios, we propose to investigate such adversarial threats on more practical forecasting models whose predictions are based on more precise features, e.g. valuations of other stock indices. Intuitively, rather than releasing adverse information to alter the sentiment about the target stock on social media, the adversaries can instead invest hence change the valuation adversely

---

∗T. N. Hoang contributed to this paper while working at Amazon.

for a selected subset of stock indices (not including the target stock) which is arguably harder to detect. Interestingly, despite being seemingly plausible given the vast literature on adversarial attack for classification models, formulating such imperceptible attack under a multivariate forecasting setup is not straightforward. This is due to several differences between forecasting and classification, particularly in terms of unique characteristic of time series, e.g., multi-step predictions, correlation over multiple time series, and probabilistic predictions.

These differences open up the question of how adversarial perturbations and robustness should be defined more properly in time series setting. Although there have been a few recent studies in this direction based on randomized smoothing (Yoon et al., 2022), these approaches are all restricted to univariate forecasting where the attack has to make adverse alterations directly to the target time series. Thus, under the less studied scenario of multivariate time-series forecasting setup, it remains unclear whether the attack to a target time series can be made instead via perturbing the other correlated time series; and whether it is defensible against such adversarial threats. In particular, as illustrated above in the stock forecasting example, there are new regimes of sparse and indirect cross time series attack under multivariate time-series scenarios, which are more effective and realistic than the direct attack in univariate cases.

In order to understand whether such new regimes of attack exists and can be defended against, we raise three questions:

1. **Indirect Attack.** Can we mislead the prediction of some target time series via perturbations on the other time series?
2. **Sparse Attack.** Can such perturbations be sparse and non-deterministic to be less perceptible?
3. **Robust Defense.** Can we defend against those indirect and imperceptible attacks?

Here we summarize our technical contributions by answering the questions above:

Regarding **indirect attack**, we provide general framework of adversarial attack in multivariate time series (see Section 3.1). Then, we devise a deterministic attack (see Section 3.2) to the state-of-the-art probabilistic multivariate forecasting model. The attack changes the model's prediction on the target time series via adversely perturbing a subset of other time series. This is achieved via formulating the perturbation as solution of an optimization task with packing constraints.

Regarding **sparse attack**, we develop a non-deterministic attack (see Section 3.3) that adversely perturbs a stochastic subset of time series related to the target time series, which makes the attack less perceptible. This is achieved via a stochastic and continuous relaxation of the above packing constraint which are shown (see Section 5) to be more effective than the deterministic attack in certain cases. Moreover, unlike deterministic attack, its differentiability makes it suitable to be directly integrated as part of a differentiable defense mechanism that can be optimized via gradient descent in an end-to-end fashion, as discussed later in Section 4.2.

Regarding **robust defense**, we propose two defense mechanisms. First, we adapt randomized smoothing to the new multivariate forecasting setup with robust certificate. Second, we devise a defense mechanism (see Section 4.2) via solving a mini-max optimization task which minimizes the maximum expected damage caused by the probabilistic attack that continually updates the generation of its adverse perturbations in response to the model updates. Their effectiveness are demonstrated across extensive experiments in Section 5.

Furthermore, our experiments in Section 5.3 demonstrate that attacks designed for univariate cases cannot be reused as an effective attack to multivariate forecasting models, which highlights the importance and novelty of our studies. The code to reproduce our experiments results can be found at `https://github.com/awslabs/gluonts/tree/dev/src/gluonts/nursery/robust-mts-attack`.

## 2 RELATED WORK

**Deep Forecasting Models.** The recent decades have witnessed a tremendous progress in DNN-based forecasting models. Given the temporal dependency of time series data, RNN and CNN-based architectures have been proved a success for time series forecasting tasks, see Rangapuram et al. (2018); Lim et al. (2020); Wang et al. (2019); Salinas et al. (2020) and Oord et al. (2016); Bai et al.

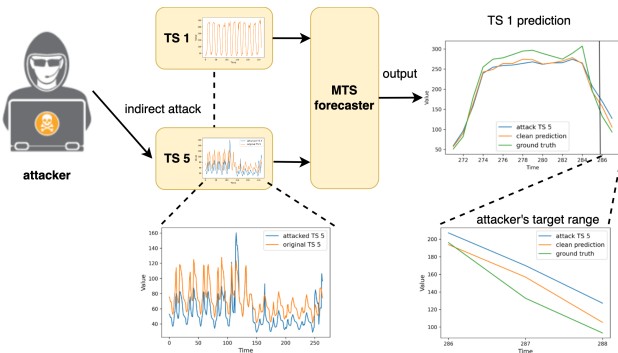

**Figure 1:** Illustration figure: an attacker misleads prediction of time series (TS) 1 at time 288 by indirectly attacking TS 5. Left plot of is authentic (orange) and perturbed (blue) versions of TS 5; right plot is no-attack (orange) and under-attack (blue) predictions for TS 1. Ground truth (green) is also plotted for comparison. No alteration is made to TS 1 but the prediction of TS 1 at the attack time step ($t = 288$) is adversely altered in the under-attack (blue) setting, which can set the prediction of TS 1 significantly away from the ground truth.

(2018) respectively. To model the uncertainty, various probabilistic models have been proposed from distributional outputs (Salinas et al., 2020; de Bézenac et al., 2020; Rangapuram et al., 2018) to distribution-free quantile-based outputs (Park et al., 2022; Gasthaus et al., 2019; Kan et al., 2022). In multivariate cases, Salinas et al. (2019) generalized DeepAR (Salinas et al., 2020) to multivariate cases and adopted low-rank Gaussian copula process to tackle the high-dimensionality challenge.

**Adversarial Attack.** Despite its success in various tasks, deep neural network is especially vulnerable to adversarial attacks (Szegedy et al., 2013) in the sense that even imperceptible adversarial noise can lead to completely different prediction. In computer vision, many adversarial attack schemes have been proposed. See Goodfellow et al. (2014b); Madry et al. (2018) for attacking image classifiers and Dai et al. (2018) for attacking graph structured data. In the field of time series, there is much less literature and even so, most existing studies on adversarial robustness of MTS models (Mode & Hoque, 2020; Harford et al., 2020) are restricted to regression and classification settings. Alternatively, Yoon et al. (2022) studied both adversarial attacks to probabilistic forecasting models, which is only restricted to univariate settings.

**Adversarial Robustness and Certification.** Against adversarial attacks, an extensive body of work has been devoted to quantifying model robustness and defense mechanisms. For instance, Fast-Lin/Fast-Lip (Weng et al., 2018) recursively computes local Lipschitz constant of a neural network; PROVEN (Weng et al., 2019) certifies robustness in a probabilistic approach. Recently, randomized smoothing has gained increasing popularity as to enhance model robustness, which was proposed by Cohen et al. (2019); Li et al. (2019) as a defense approach with certification guarantee. To the time series setting, Yoon et al. (2022) adopted randomized smoothing technique to univariate forecasting models and developed theory therein. However, we are not aware of any prior works on randomized smoothing for multivariate probabilistic models.

## 3 ADVERSARIAL ATTACK STRATEGIES

We provide a generic framework of sparse and indirect adversarial attack under a multivariate setting in Section 3.1. Then, a deterministic one to this task is introduced next in Section 3.2, followed by a stochastic attack derived in Section 3.3.

**Notations.** Denote $d$-dimensional multivariate time series $\mathbf{x}_t \in \mathbb{R}^d$ at time $t$ with its observation of $i$-th time series $x_{i,t} = [\mathbf{x}_t]_i$. We denote $\mathbf{x} = \{\mathbf{x}_t\}_{t=1}^T \in \mathbb{R}^{d \times T}$ and $\mathbf{z} = \{\mathbf{x}_{T+t}\}_{t=1}^\tau \in \mathbb{R}^{d \times \tau}$ as recent $T$ historical observations and next $\tau$-step of the future values respectively. Then, probabilistic forecaster $p_\theta$ with parameterzation $\theta$ takes history $\mathbf{x}$ to predict $\mathbf{z}$, i.e., $\mathbf{z} \sim p_\theta(\cdot \mid \mathbf{x})$. We denote the set $[d] = \{1, \ldots, d\}$ and $i$-th time series as $\boldsymbol{\delta}^i = ([\boldsymbol{\delta}_t]_i)_{t=1}^T$.

### 3.1 FRAMEWORK ON SPARSE AND INDIRECT ADVERSARIAL ATTACK

Following the notation convention of Dang-Nhu et al. (2020), given an adversarial prediction target $\mathbf{t}_{\text{adv}}$ and historical input $\mathbf{x}$ to the forecaster $p_\theta(\mathbf{z}|\mathbf{x})$, we design a perturbation matrix $\boldsymbol{\delta}$ such that

the perturbed input $\mathbf{x} + \boldsymbol{\delta}$ disturbs a statistic $\chi(\mathbf{z})$ as close as possible to $\mathbf{t}_{\mathrm{adv}}$. That is, we find $\boldsymbol{\delta}$ such that the distance between $\mathbb{E}_{\mathbf{z}|\mathbf{x}+\boldsymbol{\delta}}[\chi(\mathbf{z})]$ and $\mathbf{t}_{\mathrm{adv}}$ is minimized. Here, $\chi(\mathbf{z})$ and $\mathbf{t}_{\mathrm{adv}}$ are any arbitrary function of interest or adversarial target values with the same dimension. We focus on scenarios where the perturbed prediction is far way from original prediction by properly choosing $\chi(\cdot)$ and $\mathbf{t}_{\mathrm{adv}}$. For example, by choosing $\chi(\mathbf{z}) = \mathbf{z}$ and $\mathbf{t}_{\mathrm{adv}} = 100\mathbf{z}$, the adversary's target is to design an attack that can increase the prediction by 100 times.

Thus, suppose the adversaries want to mislead the forecasting of time series in a subset $\mathcal{I} \subset [d]$, denoted as $\mathbf{z}^{\mathcal{I}}$. Let $\chi$ be a statistic function of interest that concerns only time series in $\mathcal{I}$, i.e. $\chi(\mathbf{z}) = \chi(\mathbf{z}^{\mathcal{I}})$. To make the attack less perceptible, we impose the following sparse and indirect constraints: First, perturbation $\boldsymbol{\delta}$ cannot be direct to target time series in $\mathcal{I}$ and can be indirectly applied to a small subset of $\mathcal{I}^c = [d] \setminus \mathcal{I}$. In other words, we restrict $\boldsymbol{\delta}^{\mathcal{I}} = \mathbf{0}$ and $s(\boldsymbol{\delta}) = |\{i \in \mathcal{I}^c : \boldsymbol{\delta}^i \neq \mathbf{0}\}| \leq \kappa$ with sparsity level $\kappa \leq d$. Lastly, to avoid outlier detection, we also cap the energy of the attack such that the value of the perturbation at any coordinates is no more than a pre-defined threshold $\eta$. To sum up, the sparse and indirect attack $\boldsymbol{\delta}$ can be found via solving

$$\underset{\boldsymbol{\delta} \in \mathbb{R}^{T \times d}}{\text{minimize}} \quad \left\{ F(\boldsymbol{\delta}) \triangleq \left\| \mathbb{E}_{p_\theta(\mathbf{z}|\mathbf{x}+\boldsymbol{\delta})}\left[\chi(\mathbf{z})\right] - \mathbf{t}_{\mathrm{adv}} \right\|_2^2 \right\} \tag{3.1}$$
$$\text{subject to} \quad \|\boldsymbol{\delta}\|_{\max} \leq \eta, \; s(\boldsymbol{\delta}) \leq \kappa, \; \boldsymbol{\delta}^{\mathcal{I}} = \mathbf{0},$$

where $\|\boldsymbol{\delta}\|_{\max} = \max_{t,i} |[\boldsymbol{\delta}_t]_i|$ is the element-wise maximum norm. As such, small values of $\kappa$ and $\eta$ imply a less perceptible attack. However, solving this is intractable due to the discrete cardinality constraint on $s(\boldsymbol{\delta})$. To sidestep this, we develop two approximations in the subsequent sections which correspond to our deterministic and non-deterministic attack strategies.

## 3.2 DETERMINISTIC ATTACK

Here we present an approximated solution, inspired by the ideas in Croce & Hein (2019). We first get an intermediate solution $\hat{\boldsymbol{\delta}}$ through projected gradient descent (PGD) until it converges,

$$\hat{\boldsymbol{\delta}} \quad \leftarrow \quad \prod_{\mathcal{B}_\infty(0,\eta)} \left( \hat{\boldsymbol{\delta}} - \alpha \nabla_{\boldsymbol{\delta}} F\left(\hat{\boldsymbol{\delta}}\right) \right), \tag{3.2}$$

where $\alpha \geq 0$ is a step size and $\prod_{\mathcal{B}_\infty(0,\eta)}$ is the projection onto the $\ell_\infty$-norm ball with radius $\eta$, allowing a simple element-wise clipping: $\prod_{\mathcal{B}_\infty(0,\eta)}([\hat{\boldsymbol{\delta}}_t]_i) = \mathrm{sign}([\hat{\boldsymbol{\delta}}_t]_i)\,\eta$ if $|[\hat{\boldsymbol{\delta}}_t]_i| > \eta$ else $[\hat{\boldsymbol{\delta}}_t]_i$. With this intermediate non-sparse $\hat{\boldsymbol{\delta}}$, we retrieve for final sparse perturbation $\boldsymbol{\delta}$ via solving

$$\underset{\boldsymbol{\delta} \in \mathbb{R}^{T \times d}}{\text{minimize}} \quad \|\boldsymbol{\delta} - \hat{\boldsymbol{\delta}}\|_{\mathrm{F}} \quad \text{subject to} \quad s(\boldsymbol{\delta}) \leq \kappa, \; \boldsymbol{\delta}^{\mathcal{I}} = \mathbf{0}. \tag{3.3}$$

It turns out (3.3) can be solved analytically. Given $\hat{\boldsymbol{\delta}}$, we compute the absolute perturbation added to each row $i$, $p_i = \sum_{t=1}^{T} |[\hat{\boldsymbol{\delta}}_t]_i|$ for $i \in [d] \setminus \mathcal{I}$ and sort them in descending order $\pi$: $p_{\pi_1} \geq \cdots \geq p_{\pi_d}$. Finally, we construct the solution as $\boldsymbol{\delta}$ with $\boldsymbol{\delta}^{\pi_i} = \hat{\boldsymbol{\delta}}^{\pi_i}$ if $i \leq \kappa$ else $\mathbf{0}$.

**Remark.** $\nabla_{\boldsymbol{\delta}} F(\boldsymbol{\delta})$ involves the computation of the gradient of an expectation, which doesn't have a closed-form solution. To overcome this intractability, we adopt the re-parameterized sampling approach used in Dang-Nhu et al. (2020) and Yoon et al. (2022).

## 3.3 PROBABILISTIC ATTACK

To make the attack even less perceptible, we further show in this section an alternative approximation that results in a probabilistic sparse attack, which makes adverse alterations to a non-deterministic set of coordinates (i.e., time series and time steps). As shown in our experiment, this non-determinism appears to make the attack stronger and harder to detect.

To achieve this, we view the sparse attack vector as a random vector drawn from a distribution with differentiable parameterization. The core challenge is how to configure such a distribution whose support is guaranteed to be within the space of sparse vectors. To achieve this, we propose sparse layer, a distributional output, of a normal standard and a Dirac density combination. The output of this layer satisfied relaxed sparse support condition (see Theorem 3.2).

---

**Algorithm 1** Deterministic Adversarial Attack

---

    **input:** pre-trained model $p_\theta(\mathbf{z} \mid \mathbf{x})$, observation $\mathbf{x}$ and other parameters:
- statistic $\chi(\cdot)$, adversarial target $\mathbf{t}_{\mathrm{adv}}$, target set $\mathcal{I} \subset [d]$
- attack energy $\eta$, sparse constraint $\kappa$, PGD iterations $n$ and step size $\alpha \geq 0$

    **output:** perturbation matrix $\boldsymbol{\delta} \in \mathbb{R}^{T \times d}$ s.t. $\|\boldsymbol{\delta}\|_{\mathrm{max}} \leq \eta,\ s(\boldsymbol{\delta}) \leq \kappa,\ \boldsymbol{\delta}^{\mathcal{I}} = \mathbf{0}$
    1. initialize $\boldsymbol{\delta} = \mathbf{0}$
    **for** iteration $1, 2, \ldots, n$ **do**
        2. compute the expected loss $F(\boldsymbol{\delta})$ using Eq. (3.1)
        3. update $\boldsymbol{\delta}$ via PGD in Eq. (3.2)
    **end for**
    4. for $i \notin \mathcal{I}$, compute $p_i = \sum_{t=1}^{T} |[\boldsymbol{\delta}_t]_i|$
    5. sort $p_i$ in a descending order $\pi = (\pi_1, \ldots, \pi_d)$: $p_{\pi_1} \geq p_{\pi_2} \geq \cdots \geq p_{\pi_d}$.
    6. set $\boldsymbol{\delta}^{\pi_{\kappa+1}} = \boldsymbol{\delta}^{\pi_{\kappa+2}} = \cdots = \boldsymbol{\delta}^{\pi_d} = \mathbf{0}$ and $\boldsymbol{\delta}^{\mathcal{I}} = \mathbf{0}$. Return $\boldsymbol{\delta}$.

---

**Sparse Layer.** A sparse layer is defined as a distributional output $q(\boldsymbol{\delta}|\mathbf{x}; \beta, \gamma)$ of $\boldsymbol{\delta}$ having independent rows, such that its sample (probabilistic attack) $\boldsymbol{\delta} \sim q(\boldsymbol{\delta}|\mathbf{x}; \beta, \gamma) = \prod_i q_i(\boldsymbol{\delta}^i|\mathbf{x}; \beta, \gamma)$ satisfies sparse condition $\mathbb{E}[s(\boldsymbol{\delta})] \leq \kappa$ and $\boldsymbol{\delta}^{\mathcal{I}} = \mathbf{0}$. With $\boldsymbol{\delta}^i$ denoted as the $i$-th row (time series) of $\boldsymbol{\delta}$ and sparsity level $\kappa$, each factor distribution $q_i(\boldsymbol{\delta}^i|\mathbf{x}; \beta, \gamma)$ parameterized by $\beta$ and $\gamma$ is defined as

$$q_i\left(\boldsymbol{\delta}^i \mid \mathbf{x}; \beta, \gamma\right) \triangleq r_i(\gamma) \cdot q_i'\left(\boldsymbol{\delta}^i \mid \mathbf{x}; \beta\right) + \left(1 - r_i(\gamma)\right) \cdot D\left(\boldsymbol{\delta}^i\right), \quad (3.4)$$

where $r_i(\gamma) \triangleq \kappa \gamma_i^{\frac{1}{2}} \cdot (\sum_{i=1}^d \gamma_i)^{-\frac{1}{2}}/\sqrt{d}$, $D(\boldsymbol{\delta}^i) = \mathbb{I}(\boldsymbol{\delta}^i = \mathbf{0})$ is the Dirac density, and $q_i'(\boldsymbol{\delta}^i \mid \mathbf{x}; \beta)$ is a Gaussian $\mathbb{N}(\mu(\mathbf{x}; \beta), \sigma^2(\mathbf{x}; \beta))$.

The combination weight $r_i(\gamma)$ denotes the probability mass of the event $\boldsymbol{\delta}^i = \mathbf{0}$, which is parameterized by $\gamma$. Intuitively, this means the choice of $\{r_i(\gamma)\}_{i=1}^d$ controls the row sparsity of the random matrix $\boldsymbol{\delta}$, which can be calibrated to enforce that $\mathbb{E}[s(\boldsymbol{\delta})] \leq \kappa$. We will show in Theorem 3.1 how samples can be drawn from the combined density in (3.4).

**Theorem 3.1.** *Let $\boldsymbol{\delta}^{i'} \sim q_i'(\cdot \mid \mathbf{x}; \beta, \gamma)$ and $u_i \sim \mathbb{N}(0, 1)$ for $i = 1, \ldots, d$. Define $\boldsymbol{\delta}^i = \boldsymbol{\delta}^{i'} \cdot \mathbb{I}(u_i \leq \Phi^{-1}(r_i(\gamma)))$. Then, $\boldsymbol{\delta}^i \sim q_i(\boldsymbol{\delta}^i \mid \mathbf{x}; \beta, \gamma)$.*

Here, $q_i(\cdot|\mathbf{x}; \beta, \gamma)$ is given in (3.4) and $\Phi^{-1}$ is the inverse cumulative of the standard normal distribution. We provide the proof in the appendix.

For implementation: Let $q_i'(\cdot \mid \mathbf{x}; \beta)$ be a distribution over dense vectors, e.g. $\mathbb{N}(\mu(\beta), \sigma^2(\beta)\mathbf{I})$, and $u_i \sim \mathbb{N}(0, 1)$ for $i \in [d]$. We can construct a binary mask $m_i = \mathbb{I}(u_i \leq \Phi^{-1}(r_i(\gamma)))$, $i \in [d]$, where $r_i(\gamma)$ is defined above. Next, for each $i \in [d]$, we draw $\boldsymbol{\delta}^{i'}$ from $q_i'(\cdot \mid \mathbf{x}; \beta)$ and obtain $\boldsymbol{\delta}^i$ by $\boldsymbol{\delta}^i = \boldsymbol{\delta}^{i'} \cdot m_i$ where $\cdot$ denotes the element-wise multiplication. Finally, we set $\boldsymbol{\delta}^{\mathcal{I}} = \mathbf{0}$.

Theorem 3.2 proves that $\boldsymbol{\delta}$ sampled from (3.4) would meet the constraint $\mathbb{E}[s(\boldsymbol{\delta})] \leq \kappa$. Put together, Theorem 3.1 and Theorem 3.2 enable differentiable optimization of a sparse attack as desired.

**Theorem 3.2.** *Let $\boldsymbol{\delta} \sim q(\cdot \mid \mathbf{x}; \beta, \gamma)$. Then, $\mathbb{E}[s(\boldsymbol{\delta})] \leq \kappa$.*

**Remark.** Note that we can also obtain a direct sparse constraint on $s(\boldsymbol{\delta})$ by applying Theorem 3.2 to a smaller quantity $c\kappa$ for $c \in (0, 1)$. Then, by the Markov inequality, with probability at least $1 - c$, we have $s(\boldsymbol{\delta}) \leq \mathbb{E}[s(\boldsymbol{\delta})]/c = c\kappa/c = \kappa$. We provide the proof of Theorem 3.2 in Appendix C.

**Optimizing Sparse Layer.** The differentiable parameterization of the above sparse layer can therefore be optimized for maximum attack impact via minimizing the expected distance between the attacked statistic and adversarial target:

$$\min_{\beta, \gamma} H(\beta, \gamma) \triangleq \mathbb{E}_{\boldsymbol{\delta} \sim q(.|\mathbf{x}; \beta, \gamma)} \left\| \mathbb{E}_{\mathbf{z} \sim p_\theta(\mathbf{z}|\mathbf{x}+\boldsymbol{\delta})}\left[\chi(\mathbf{z})\right] - \mathbf{t}_{\mathrm{adv}} \right\|_2^2. \quad (3.5)$$

This attack is probabilistic in two ways. First, the magnitude of the perturbation $\delta$ is a random variable from distribution $q(\cdot \mid \mathbf{x})$. Second, the non-zero components of the mask depend on the random Gaussian samples, which brings another degree of non-determinism into the design, making the attack less perceptible and harder to detect. See Algorithm 4 in Appendix A for the implementation.

**Remark.** There are three important advantages of the above probabilistic sparse attack. First, by viewing the attack vector as random variable drawn from a learnable distribution instead of fixed parameter to be optimized, we are able to avoid solving the NP-hard problem (3.1) as usually approached in previous literature (Croce & Hein, 2019). Second, our approach introduces multiple degree of non-determinism to the attack vector, apparently making it more stealth and powerful (see Section 5). Last, unlike the deterministic attack which has two separate, decoupled approximation stages that cannot be optimized end-to-end due to the non-convex and non-differentiable constraint in (3.1), the probabilistic attack model is entirely differentiable. Therefore, it can be directly integrated as part of a differentiable defense mechanism that can be optimized via gradient descent in an end-to-end fashion – see Section 4.2 for more details. Again, we adopt re-parameterizaed sampling approach to compute the gradient of the expectation in Eq. (3.5).

## 4 DEFENSE MECHANISMS AGAINST ADVERSARIAL ATTACKS

The adversarial attack on probabilistic forecasting models was investigated under the univariate time series setting (Dang-Nhu et al., 2020; Yoon et al., 2022). Beyond basic data augmentation (Wen et al., 2020), we develop more effective defense mechanism to enhance model robustness via randomized smoothing (in Section 4.1) and mini-max defense using sparse layer (in Section 4.2).

### 4.1 RANDOMIZED SMOOTHING DEFENSE

Randomized smoothing (RS) (Cohen et al., 2019) is a post-training defense technique. Having never been considered to multivariate setting to the best of our knowledge, we apply RS to our multivariate forecasters $z(\mathbf{x}) \sim p_\theta(\mathbf{z} \mid \mathbf{x})$ which maps $\mathbf{x}$ to a random vector $z(\mathbf{x})$ distributed by $p_\theta(\mathbf{z} \mid \mathbf{x})$. Let $\mathbb{P}_z(z(\mathbf{x}) \preceq \mathbf{r})$ denote the CDF of such random outcome vector where $\preceq$ denotes the element-wise inequality, the RS version

$$g_\sigma(\mathbf{x}) \quad = \quad \mathbb{E}_\epsilon \Big[ z(\mathbf{x} + \boldsymbol{\epsilon}) \Big] \tag{4.1}$$

of $z(\mathbf{x})$ with noise level $\sigma > 0$ and $\boldsymbol{\epsilon} \sim \mathbb{N}(0, \sigma^2 \mathbf{I})$ is a random vector whose CDF is defined as

$$\mathbb{P}_{g_\sigma} \Big( g_\sigma(\mathbf{x}) \preceq \mathbf{r} \Big) \quad \triangleq \quad \mathbb{E}_{\boldsymbol{\epsilon} \sim \mathbb{N}(\mathbf{0}, \sigma^2 \mathbf{I})} \Big[ \mathbb{P}_z \Big( z(\mathbf{x} + \boldsymbol{\epsilon}) \preceq \mathbf{r} \Big) \Big] \tag{4.2}$$

where we abuse the notation $\boldsymbol{\epsilon} \sim \mathbb{N}(0, \sigma^2 \mathbf{I})$ to indicate the (scalar) entries of the matrix $\boldsymbol{\epsilon}$ are independently and identically distributed by $\mathbb{N}(0, \sigma^2)$. Computing the output of the smoothed forecaster $g_\sigma(\mathbf{x})$ is intractable in general since the integration of $z(\mathbf{x} + \boldsymbol{\epsilon})$ with $\mathbb{N}(0, \sigma^2 \mathbf{I})$ cannot be done analytically. However, it can still be approximated with arbitrarily high accuracy via MC sampling with a sufficiently large number of samples. Check Algorithm 2 for a detailed implementation.

---

**Algorithm 2** Randomized Smoothing

**input:** forecaster $z(\mathbf{x}) \sim p_\theta(\mathbf{z}|\mathbf{x})$ and:
- number of samples $n$
- noise level $\sigma$
- input $\mathbf{x}$

**output:** $g_\sigma(\mathbf{x})$
initialize $g_\sigma(\mathbf{x}) = 0$
**for** $e = 1, 2, \ldots, n$ **do**
    1. sample $[\boldsymbol{\epsilon}_t^{(e)}]_i \sim \mathbb{N}(0, \sigma^2) \; \forall (t, i)$
    2. $g_\sigma(\mathbf{x}) \leftarrow g_\sigma(\mathbf{x}) + (1/n) \cdot z(\mathbf{x} + \boldsymbol{\epsilon}^{(e)})$
**end for**

**Algorithm 3** Mini-max Defense

**input:** dataset $\mathcal{D}$ of $(\mathbf{x}, \mathbf{z})$ pairs and parameters:
- sparse constraint $\kappa$ for $q$ in Eq. (4.6)
- number of optimization iterations $n$

**output:** forecasting model $z(\mathbf{x}) \sim p_\theta(\mathbf{z} \mid \mathbf{x})$.
**for** $e = 1, 2, \ldots, n$ **do**
    1. Fix $\theta$, minimize $-\sum_{(\mathbf{x}, \mathbf{z}) \sim \mathcal{D}} \ell_g(\phi; \mathbf{x}, \mathbf{z}, \theta)$
with respect to $\phi$ – see Eq. (4.5)
    2. Fix $\phi$, maximize $\sum_{(\mathbf{x}, \mathbf{z}) \sim \mathcal{D}} \ell_p(\theta; \mathbf{x}, \mathbf{z}, \phi)$
with respect to $\theta$ – see Eq. (4.6)
**end for**

---

For the randomized smoothing version $g_\sigma$ of the base forecaster $z(\mathbf{x}) \sim p_\theta(\mathbf{z}|\mathbf{x})$, we establish a robustness guarantee or certificate in the following theorem.

**Theorem 4.1** (Robust Certificate). *Given an input $\mathbf{x}$, let $g_\sigma(\mathbf{x})$ be defined in Eq. (4.1). Let $G(\mathbf{r}) = \mathbb{P}_{g_\sigma}(g_\sigma(\mathbf{x}) \preceq \mathbf{r})$ and $G_{\boldsymbol{\delta}}(\mathbf{r}) = \mathbb{P}_{g_\sigma}(g_\sigma(\mathbf{x} + \boldsymbol{\delta}) \preceq \mathbf{r})$. For any $\boldsymbol{\delta}$, we have*

$$\sup_{\mathbf{r} \in \mathbb{R}^d} \Big| G(\mathbf{r}) - G_{\boldsymbol{\delta}}(\mathbf{r}) \Big| \quad \leq \quad \frac{\sqrt{d}}{\sigma} \cdot \Big\| \boldsymbol{\delta} \Big\|_{\mathrm{F}} . \tag{4.3}$$

This shows that the difference between the CDFs of the smoothed forecaster on authentic and perturbed input, i.e. $g_\sigma(\mathbf{x})$ and $g_\sigma(\mathbf{x} + \boldsymbol{\delta})$, is guaranteed to be no more than $O(\|\boldsymbol{\delta}\|_F)$. We defer the formal proof to Appendix C.

**Remark.** Different from Theorem 1 in Yoon et al. (2022) that only applies to univariate cases, our Theorem 4.1 provides a more general robustness guarantee as it's available for multivariate setting. Also, Theorem 1 in Yoon et al. (2022) only holds for $\boldsymbol{\delta} \to 0$, but our Theorem 4.1 holds for any $\boldsymbol{\delta}$.

## 4.2 MINI-MAX DEFENSE

As discussed in Section 3.3, the sparse layer is differentiable, which is suitable to be directly integrated as part of a differentiable defense mechanism that can be optimized via gradient descent in an end-to-end fashion. To fix the idea, with a sparse layer $q(\cdot \mid \mathbf{x}; \phi)$ having parameters $\phi = (\beta, \gamma)$ in Eq. (3.4), we propose to train the forecaster by minimizing the worst-case loss caused by $q(\cdot \mid \mathbf{x}; \phi)$:

$$\min_\phi \max_\theta \sum_{(\mathbf{x}, \mathbf{z}) \sim \mathcal{D}} \Big[ \ell_p(\theta; \mathbf{x}, \mathbf{z}, \phi) - \ell_g(\phi; \mathbf{x}, \mathbf{z}, \theta) \Big]. \tag{4.4}$$

Here $\ell_g(\phi; \mathbf{x}, \mathbf{z}, \theta)$ is a function of $\phi$ conditioned on $(\mathbf{x}, \mathbf{z}, \theta)$ while $\ell_p(\theta; \mathbf{x}, \mathbf{z}, \phi)$ is a function of $\theta$ conditioned on $(\mathbf{x}, \mathbf{z}, \phi)$ as follows

$$\ell_g(\phi; \mathbf{x}, \mathbf{z}, \theta) \triangleq \mathbb{E}_{q(\boldsymbol{\delta}|\mathbf{x};\phi)} \left[ \mathbb{E}_{p_\theta(\mathbf{z}'|\mathbf{x}+\boldsymbol{\delta})} \Big\| \mathbf{z}' - \mathbf{z} \Big\|^2 \right] \tag{4.5}$$

$$\ell_p(\theta; \mathbf{x}, \mathbf{z}, \phi) \triangleq \mathbb{E}_{q(\boldsymbol{\delta}|\mathbf{x};\phi)} \Big[ \log p_\theta \big( \mathbf{z} \mid \mathbf{x} + \boldsymbol{\delta} \big) \Big] \tag{4.6}$$

where the expectation is taken over $\boldsymbol{\delta} \sim q(\boldsymbol{\delta}|x; \phi)$ with $q$ given by Eq. (3.4), each pair $(\mathbf{x}, \mathbf{z})$ represents a training data point in our dataset with $\mathbf{x} = \{\mathbf{x}_t\}_{t=1}^T$ and $\mathbf{z} = \{\mathbf{x}_{T+t}\}_{t=1}^\tau$.

Solving Eq. (4.4) therefore means finding a stable state where the model parameter is conditioned to perform best in the worst situation where the adversarial noises are also conditioned to generate the most impact even in the most benign scenario. This can be achieved by alternating between (1) minimizing $-\ell_g$ in Eq. (4.5) with respect to $(\beta, \gamma)$ and (2) maximizing $\ell_p$ in Eq. (4.6) with respect to $\theta$. We call this defense mechanism a mini-max defense. We note that similar ideas have been previously exploited in deep generative models, such as GAN (Goodfellow et al., 2014a) and WGAN (Arjovsky et al., 2017). See Algorithm 3 for a detailed description.

**Remark.** Unlike the sparse layer used in attack, the sparse layer used to simulate mock attack in our defense strategy does not have access to the actual attack sparsity parameter $\kappa$ or the set of target time series $\mathcal{I}$. Hence, we need to set the sparsity $\kappa$ as a tuning parameter and skip the last step of the sparse layer described in Section 3.3 where we set $\boldsymbol{\delta}^\mathcal{I} = \mathbf{0}$.

## 5 EXPERIMENTS

We conduct numerical experiments to demonstrate the effectiveness of our proposed indirect sparse attack on a multivariate probabilistic forecasting models and compare various defense mechanisms.

## 5.1 EXPERIMENT SETUPS

**Dataset.** We include Traffic (Asuncion & Newman, 2007), Electricity (Asuncion & Newman, 2007), Taxi (Taxi & Commission, 2015), Wiki (Lai, 2017). See Appendix B.1 for more information.

**Multivariate Forecaster.** We consider DeepVAR Salinas et al. (2020) which is state-of-the-art multivariate probabilistic models with implementation available pytorch-ts (Rasul, 2021) with target dimension 10. For more details on the model parameters, see Appendix B.2.

**Data Augmentation (DA) and Randomized Smoothing (RS).** Following the convention in Dang-Nhu et al. (2020); Yoon et al. (2022), we use relative noises in both data augmentation and randomized smoothing. That is, given a sequence of observation $\mathbf{x} = ([\mathbf{x}_t]_i)_{i,t} \in \mathbb{R}^{d \times T}$, we draw i.i.d. noise samples $[\boldsymbol{\epsilon}_t]_i \sim \mathbb{N}(0, \sigma^2)$ and produce noisy input as $[\tilde{\mathbf{x}}_t]_i \leftarrow [\mathbf{x}_t]_i (1 + [\mathbf{x}_t]_i)$. In data augmentation, we train model with noisy input $[\tilde{\mathbf{x}}_t]_i$. In RS, the base model is still trained on noisy input $[\tilde{\mathbf{x}}_t]_i$ with noise level $\sigma$. The noise level $\sigma$ remains the same across DA and RS.

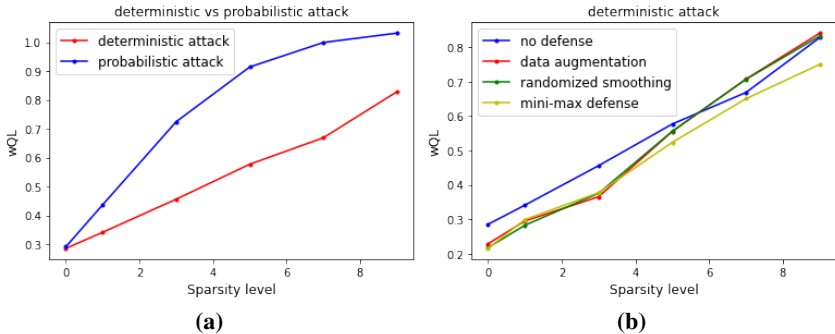

**Figure 2:** Plots of (a) averaged wQL under sparse indirect attack against the sparsity level on electricity dataset. The underlying model is a clean DeepVAR without defense. Target time series $\mathcal{I} = \{1\}$ and attacked time stamp $H = \{\tau\}$; and (b) & (c) averaged wQL under different defense mechanisms on electricity dataset for deterministic & probabilistic attack respectively.

**Metrics.** We adopt weighted quantile loss (wQL) to measure the performance. (See Appendix B.3.)

## 5.2 EXPERIMENT RESULTS

**Traffic, Electricity, and More Datasets.** Averaged wQL loss is reported in Table 1 and Table 2 for Traffic and Electricity dataset respectively. The attacks include both deterministic and probabilistic ones for both single and multiple target time series and time horizons. Besides, we plot wQL under both attacks against sparsity level to better visualize the effect of different types of attack. See Figure 2. More experiment results with error bars on additional datasets can be found in Appendix B.4.

**Message 1: Sparse, Indirect Attack is effective, and becomes more effective as $\kappa$ increases.** In the experiment, we can verify the effectiveness of sparse indirect attack, that is, one can attack the prediction of one time series without directly attacking the history of this time series. For example in Table 2, under deterministic attack, the average wQL is increased by 20% by only attacking one out of nine remaining time series (there are totally 10 but the target time series is excluded).

Moreover, attacking half of the time series can increase average wQL by 102%! This observation is even more noticeable under probabilistic attack: average wQL can be increased by 215% with 50% of the time series attacked. Besides, wQL loss increases as $\kappa$ increases, which is also an evidence that sparse indirect attack is effective.

**Message 2: Probabilistic Attack is more effective than Deterministic Attack, especially for small $\kappa$.** In general, average wQL increases as $\kappa$ increases and probabilistic attack appears to be more effective than deterministic one, see Figure 4a and Table 2. For example, under no defense when $\kappa = 7$, probabilistic attack causes 50% larger wQL loss than deterministic one.

**Message 3: Randomized Smoothing (RS) and Mini-Max are more robust than Data Augmentation (DA).** As can be seen in Figure 4b, Table 2 and Table 1, all three defense methods can bring robustness to the forecasting model. Data augmentation and randomized smoothing works well under small $\kappa$ and mini-max defense achieves comparable performance as data augmentation and randomized smoothing under small $\kappa$ and outperforms them under large $\kappa$.

**Table 1:** Average wQL on **Traffic** dataset under **deterministic** and **probabilistic** attack. Target time series $\mathcal{I} = \{1, 5\}$ and attacked time stamp $H = \{\tau - 1, \tau\}$. Smaller is better.

| | deterministic attack | | | | probabilistic attack | | | |
|---|---|---|---|---|---|---|---|---|
| sparsity ($\kappa$) | no defense | DA | RS | mini-max | no defense | DA | RS | mini-max |
| no attack | 0.2283 | 0.1573 | **0.1529** | 0.1837 | 0.2283 | 0.1573 | **0.1529** | 0.1837 |
| 1 | 0.2190 | 0.1543 | **0.1529** | 0.1701 | 0.2428 | 0.1807 | **0.1796** | 0.1904 |
| 3 | 0.2150 | 0.1884 | 0.1890 | **0.1687** | 0.2219 | 0.2564 | 0.2467 | **0.1714** |
| 5 | 0.2772 | 0.2729 | 0.2648 | **0.1688** | 0.2719 | 0.3026 | 0.3003 | **0.1883** |
| 7 | 0.3620 | 0.3597 | 0.3535 | **0.1779** | 0.3529 | 0.2893 | 0.2824 | **0.1846** |
| 9 | 0.4635 | 0.4058 | 0.4240 | **0.1970** | 0.4075 | 0.3544 | 0.3376 | **0.1911** |

**Table 2:** Average wQL on **Electricity** dataset under **deterministic** and **probabilistic** attack. Target time series $\mathcal{I} = \{1\}$ and attacked time stamp $H = \{\tau\}$. Smaller is better.

| sparsity ($\kappa$) | deterministic attack | | | | probabilistic attack | | | |
|---|---|---|---|---|---|---|---|---|
| | no defense | DA | RS | mini-max | no defense | DA | RS | mini-max |
| no attack | 0.2853 | 0.2288 | 0.2176 | **0.2154** | 0.2909 | 0.2374 | **0.2237** | 0.2342 |
| 1 | 0.3410 | 0.2949 | **0.2826** | 0.2990 | **0.4364** | 0.5923 | 0.5940 | 0.4935 |
| 3 | 0.4559 | **0.3655** | 0.3757 | 0.3775 | 0.7245 | 0.5738 | **0.4581** | 0.8079 |
| 5 | 0.5770 | 0.5554 | 0.5560 | **0.5273** | 0.9143 | 0.8422 | 0.9276 | **0.5265** |
| 7 | 0.6687 | 0.7076 | 0.7072 | **0.6506** | 0.9991 | 0.8267 | 1.0100 | **0.6161** |
| 9 | 0.8282 | 0.8412 | 0.8327 | **0.7503** | 1.0317 | 0.8139 | 0.8919 | **0.6466** |

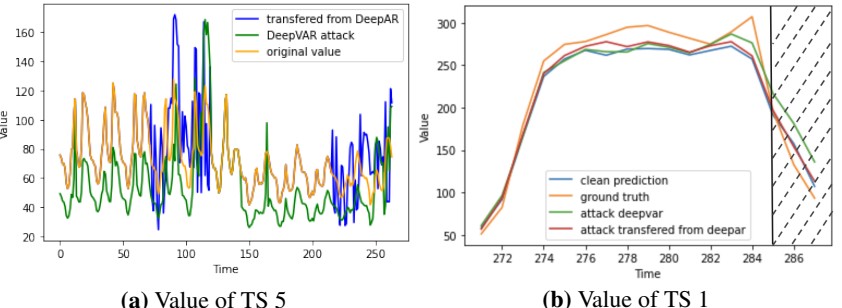

**(a)** Value of TS 5        **(b)** Value of TS 1

**Figure 3:** Plots of (a) authentic (orange), DeepAR-attacked (blue) and DeepVAR-attacked (green) versions of time-series (TS) 5; and (b) ground-truth (orange), no-attack (blue), under-DeepAR-attack (red) and under-DeepVAR-attack (green) predictions for TS 1. Shaded area is attacker's target range. Compared to clean prediction, the value of TS 1 at the attack time step ($t = 288$) were adversely altered by DeepVAR-attack (green) but only slightly altered by DeepAR-attack (red). The wQL loss under no attack: 0.288, under DeepAR attack: 0.322, under DeepVAR attack: 0.390.

### 5.3 NON-TRANSFERRABLITY BETWEEN UNIVARIATE AND MULTIVARIATE CASES

From the above Section 5.2, we verify the effectiveness of sparse indirect attack of multivariate forecasting models. In this subsection, we investigate the transferrability from univariate attack to multivariate attack. To be specific, we study the question whether the adversarial perturbation generated by univariate models can be transferred to multivariate models as an indirect attack.

In empirical experiments, we choose $\kappa = 1$ and other parameters are the same as what are described in Section 5.1. It turns out TS 5 is selected by Algorithm 1 to harm the prediction of TS 1 when $\kappa = 1$. Thus, we use the technique in Dang-Nhu et al. (2020); Yoon et al. (2022) to generate univariate attack on TS 5 from DeepAR. Note that only the history of TS 5 has been adversely altered. The attacked time series 5 is further fed into DeepVAR model.

**Experiment Result.** The averaged wQL loss is reported in Table 12 in Appendix D. For a better visualization, the history of TS 5 and prediction of TS 1 are plotted in Figure 3a and Figure 3b respectively. From the experiment results in Table 12, we observe that multivariate attack is 3x more effective than univariate attack, which is also a reason why multivariate attack worth investigation.

## 6 CONCLUSION

In this work, we investigate the existence of sparse indirect attack for multivariate time series forecasting models. We propose both deterministic approach and a novel probabilistic approach to finding effective adversarial attack. Besides, we adopt the randomized smoothing technique from image classification and univariate time series to our framework and design another mini-max optimization to effectively defend the attack delivered by our attackers. To the best of our knowledge, this is the first work to study sparse indirect attack on multivariate time series and develop corresponding defense mechanisms, which could inspire a future research direction.

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

# A    PROBABILISTIC ATTACK ALGORITHM

---

**Algorithm 4** Probabilistic Adversarial Attack

---

   **input:**  pre-trained model $p_\theta(\mathbf{z} \mid \mathbf{x})$, observation $\mathbf{x}$ and other parameters:
   - statistic $\chi(\cdot)$, adversarial target $\mathbf{t}_{\mathrm{adv}}$, target set $\mathcal{I} \subset [d]$
   - attack energy $\eta$, sparse constraint $\kappa$, number of iterations $n$

   **output:**  perturbation matrix $\boldsymbol{\delta} \in \mathbb{R}^{T \times d}$ s.t. $\|\boldsymbol{\delta}\|_{\max} \leq \eta$, $\mathbb{E}[s(\boldsymbol{\delta})] \leq \kappa$, $\boldsymbol{\delta}^{\mathcal{I}} = \mathbf{0}$
   1. randomly initialize a sparse layer $q(\cdot|\mathbf{x}; \beta, \gamma)$
   **for** iteration $1, 2, \ldots, n$ **do**
       2. compute the expected loss $H(\beta, \gamma)$ using Eq. (3.5)
       3. update $\beta, \gamma$ via first-order optimization method
   **end for**
   4. draw $\boldsymbol{\delta} \sim q(\cdot|\mathbf{x}; \beta, \gamma)$ and return

---

# B    DETAILS ON THE EXPERIMENT SETTING

## B.1    DATASETS

   - Traffic: hourly occupancy rate, between 0 and 1, of 963 San Francisco car lanes.
   - Taxi: traffic time series of New York taxi rides taken at 1214 locations for every 30 minutes from January 2015 to January 2016 and considered to be heterogeneous. We use the taxi-30min dataset provided by GluonTS.
   - Wiki: daily page views of 2000 Wikipedia pages used in Gasthaus et al. (2019).
   - Electricity: consists of hourly electricity consumption time series from 370 customers.

**Table 3:** Summary of statistics of the datasets used, including prediction length $\tau$, domain, frequency, dimension, and time steps.

| dataset | prediction length $\tau$ | domain | frequency | dimension | time steps |
|---|---|---|---|---|---|
| Traffic | 24 | $\mathbb{R}^+$ | H | 963 | 10413 |
| Taxi | 24 | $\mathbb{N}$ | 30-min | 1214 | 1488 |
| Wiki | 30 | $\mathbb{N}$ | D | 2000 | 792 |
| Electricity | 24 | $\mathbb{R}^+$ | H | 370 | 5790 |

## B.2    HYPER-PARAMETER CHOICE

**Traffic.**    We target at the prediction of $\chi(\mathbf{z}) = (x_{1,T+\tau-1}, x_{1,T+\tau}, x_{5,T+\tau-1}, x_{5,T+\tau})$. We choose prediction length $\tau = 24$ and context length $T = 4\tau = 96$, and sparsity level $\kappa = 1, 3, 5, 7, 9$.

**Wiki.**    We target at the prediction of $\chi(\mathbf{z}) = x_{1,T+\tau}$. We choose prediction length $\tau = 30$ and context length $T = 4\tau = 120$, and sparsity level $k = 1, 3, 5, 7, 9$.

**Electricity & Taxi.**    We target at the prediction of the first time series at the last prediction time step, i.e. target time series $\mathcal{I} = \{1\}$ and time horizon to attack $H = \{\tau\}$, so $\chi(\mathbf{z}) = x_{1,T+\tau}$. We choose prediction length $\tau = 24$ and context length $T = 4\tau = 96$, and sparsity level $\kappa = 1, 3, 5, 7, 9$.

For all experiments, we train a DeepVAR with rank 5. The attack energy $\eta = c_1 \max |\mathbf{x}|$, is proportional to the largest element of the past observation in magnitude, where $c_1$ is set to 0.5. For the adversarial target $\mathbf{t}_{\mathrm{adv}}$, we first draw a prediction $\hat{\mathbf{x}}$ from un-attacked model $p_\theta(\cdot|\mathbf{x})$ and choose $\mathbf{t}_{\mathrm{adv}} = c_2\hat{\mathbf{x}}$ for constants $c_2 = 0.5$ and 2.0. Following the process in Yoon et al. (2022),  we report the largest error produced by these choices of constants. Unless otherwise stated, the number of

sample paths drawn from the prediction distribution $n = 100$ to quantify quantiles $q_{i,t}^{(\alpha)}$. In minimax defense, the sparsity level of the sparse layer is set to 5 for all cases. For the noise level $\sigma$ in DA and RS, we select them via a validation set and it turns out no $\sigma$ is uniformly better than the others across different sparsity level. Thus, $\sigma = 0.1$ is chosen in the empirical evaluation. For an ablation study on the effect of $\sigma$, see Table 6 in Appendix B.4.

## B.3 METRICS

We measure the performance of model under attacks by the popular metric especially for probabilistic forecasting models: weighted quantile loss (wQL), which is defined as

$$\mathrm{wQL}(\alpha) = 2 \cdot \frac{\sum_{i,t}[\alpha \max(x_{i,t} - q_{i,t}^{(\alpha)}, 0) + (1 - \alpha)\max(q_{i,t}^{(\alpha)} - x_{i,t}, 0)]}{\sum_{i,t}|x_{i,t}|}$$

where $\alpha \in (0,1)$ is a quantile level. In practical application, under-prediction and over-prediction may cost differently, suggesting wQL should be one's main consideration especially for probabilistic forecasting models. In the subsequent sections, we calculate average wQL over a range of $\alpha = [0.1, 0.2, \ldots, 0.9]$ and evaluate the performance in terms of averaged wQL.

## B.4 MORE RESULTS

To measure the performance of a forecasting model, other metrics like Weighted Absolute Percentage Error (WAPE) or Weighted Squared Error (WSE) are also considered by a large body of literature. For completeness, we present the definition of WAPE and WSE:

$$\mathrm{WAPE} = \sum \left| \frac{\text{predicted value}}{\text{true value}} - 1 \right| = \frac{1}{|I||H|} \sum_{i \in I, h \in H} \left| \frac{\frac{1}{n}\sum_{j=1}^{n} \hat{x}_{T+h,i}^{j}}{x_{T+h,i}} - 1 \right|,$$

$$\mathrm{WSE} = \sum \left( \frac{\text{predicted value}}{\text{true value}} - 1 \right)^2 = \frac{1}{|I||H|} \sum_{i \in I, h \in H} \left( \frac{\frac{1}{n}\sum_{j=1}^{n} \hat{x}_{T+h,i}^{j}}{x_{T+h,i}} - 1 \right)^2,$$

where $\hat{x}_{i,j}$ is the predicted values from forecasting model. We report WAPE, WSE and wQL under deterministic and probabilistic attacks on electricity dataset in Table 4 and Table 5.

**Table 4:** Metrics on **Electricity** dataset under **deterministic** attack. Target time series $\mathcal{I} = \{1\}$ and attacked time stamp $H = \{\tau\}$. Smaller is better.

| Sparsity ($\kappa$) | no defense | | | data augmentation | | |
|---|---|---|---|---|---|---|
| | WAPE | WSE | wQL | WAPE | WSE | wQL |
| no attack | 0.4005±0.2036 | 0.2360±0.2525 | 0.2853±0.1684 | 0.4241±0.2092 | 0.2596±0.2625 | 0.2288±0.1497 |
| 1 | 0.4900±0.2488 | 0.3529±0.3769 | 0.3410±0.2106 | 0.4123±0.1829 | 0.2310±0.1934 | 0.2949±0.1138 |
| 3 | 0.6382±0.3434 | 0.6222±0.5886 | 0.4559±0.2917 | 0.5654±0.2475 | 0.4313±0.3707 | 0.3655±0.1876 |
| 5 | 0.7524±0.3675 | 0.8123±0.6218 | 0.5770±0.3218 | 0.7460±0.3803 | 0.8201±0.6628 | 0.5554±0.2833 |
| 7 | 0.8786±0.4171 | 1.0889±0.7785 | 0.6687±0.3702 | 0.8465±0.4014 | 1.0102±0.6389 | 0.7076±0.2985 |
| 9 | 1.0134±0.4541 | 1.4028±0.9685 | 0.8282±0.4023 | 0.9093±0.4454 | 1.1883±0.7720 | 0.8412±0.3395 |

| Sparsity ($\kappa$) | randomized smoothing | | | mini-max defense | | |
|---|---|---|---|---|---|---|
| | WAPE | WSE | wQL | WAPE | WSE | wQL |
| no attack | 0.3501±0.1630 | 0.1710±0.1486 | 0.2176±0.1068 | 0.3237±0.1379 | 0.1394±0.0913 | 0.2154±0.0917 |
| 1 | 0.4209±0.1700 | 0.2298±0.1683 | 0.2826±0.1003 | 0.4498±0.2253 | 0.2949±0.2276 | 0.2990±0.1825 |
| 3 | 0.5887±0.2543 | 0.4644±0.3784 | 0.3757±0.1797 | 0.7447±0.3758 | 0.8120±0.6684 | 0.3775±0.3358 |
| 5 | 0.7504±0.3607 | 0.8002±0.5999 | 0.5560±0.2779 | 0.9603±0.4190 | 1.2419±0.8369 | 0.5273±0.3845 |
| 7 | 0.8353±0.4315 | 1.0369±0.7496 | 0.7072±0.3152 | 1.1056±0.4847 | 1.6504±1.0591 | 0.6506±0.4350 |
| 9 | 0.9986±0.5026 | 1.4574±0.9998 | 0.8327±0.3717 | 1.2476±0.4860 | 1.9870±1.0815 | 0.7503±0.4306 |

Also, Table 6 reports the effect of choosing different values for $\sigma$ in data augmentation and randomized smoothing.

Next, we report wQL loss of Taxi and Wiki dataset under both types of attacks in Table 7, Table 8, Table 9 and Table 10 respectively.

We also report the results of attacking effects with different attacked time stamp $H$ in Table 11.

**Table 5:** Metrics on **electricity** dataset under **probabilistic** attack using sparse layer. Target time series $\mathcal{I} = \{1\}$ and attacked time stamp $H = \{\tau\}$. Smaller is better.

| Sparsity ($\kappa$) | no defense | | | data augmentation | | |
|---|---|---|---|---|---|---|
| | WAPE | WSE | wQL | WAPE | WSE | wQL |
| no attack | 0.3842±0.2620 | 0.2162±0.3044 | 0.2909±0.0748 | 0.3074±0.1746 | 0.1250±0.0946 | 0.2374±0.0764 |
| 1 | 0.6230±0.6324 | 0.7881±1.1864 | 0.4364±0.1296 | 0.7476±0.7240 | 1.0830±1.8593 | 0.5923±0.0913 |
| 3 | 1.0540±0.7522 | 1.6768±1.4810 | 0.7245±0.2434 | 0.8484±0.6809 | 1.1834±1.3998 | 0.5738±0.1759 |
| 5 | 1.2078±0.7451 | 2.0139±2.0667 | 0.9143±0.3235 | 1.1444±0.6665 | 1.7538±1.4318 | 0.8422±0.2945 |
| 7 | 1.3236±0.7310 | 2.2863±1.8336 | 0.9991±0.3505 | 1.1304±0.6522 | 1.7031±1.4053 | 0.8267±0.2823 |
| 9 | 1.3656±0.8671 | 2.6166±2.6679 | 1.0317±0.3707 | 1.0912±0.6181 | 1.5727±1.2081 | 0.8139±0.2827 |

| Sparsity ($\kappa$) | randomized smoothing | | | mini-max defense | | |
|---|---|---|---|---|---|---|
| | WAPE | WSE | wQL | WAPE | WSE | wQL |
| no attack | 0.2858±0.1547 | 0.1056±0.0761 | 0.2237±0.0750 | 0.3218±0.1429 | 0.1240±0.0830 | 0.2342±0.0710 |
| 1 | 0.7683±0.8771 | 1.3596±2.7290 | 0.5940±0.1142 | 0.6990±0.6957 | 0.9726±1.7182 | 0.4935±0.1450 |
| 3 | 0.6784±0.5230 | 0.7337±0.7698 | 0.4581±0.1301 | 0.9909±0.7564 | 1.5540±1.8925 | 0.8079±0.2838 |
| 5 | 1.2310±0.7025 | 2.0090±1.6609 | 0.9276±0.3208 | 0.6966±0.4554 | 0.6927±0.8752 | 0.5265±0.1611 |
| 7 | 1.3496±0.6777 | 2.2809±1.7240 | 1.0100±0.3554 | 0.8424±0.7803 | 1.3186±1.7286 | 0.6161±0.1986 |
| 9 | 1.1978±0.6742 | 1.8894±1.5309 | 0.8919±0.3072 | 0.8691±0.7410 | 1.3043±2.0663 | 0.6466±0.2054 |

**Table 6:** Average wQL on **Electricity** dataset under **deterministic** attack. The defense is data augmentation and randomized smoothing with varying $\sigma = 0.1, 0.2, 0.3$. Target time series $\mathcal{I} = \{1\}$ and attacked time stamp $H = \{\tau\}$. Smaller is better.

| Sparsity ($\kappa$) | no defense | $\sigma = 0.1$ | | $\sigma = 0.2$ | | $\sigma = 0.3$ | | mini-max |
|---|---|---|---|---|---|---|---|---|
| | | DA | RS | DA | RS | DA | RS | |
| no attack | 0.2853 | 0.2288 | 0.2176 | 0.2321 | 0.2389 | 0.2999 | 0.3053 | **0.2154** |
| 1 | 0.3410 | 0.2949 | 0.2826 | **0.2717** | 0.2866 | 0.2959 | 0.3456 | 0.2990 |
| 3 | 0.4559 | **0.3655** | 0.3757 | 0.4822 | 0.4421 | 0.4323 | 0.3930 | 0.3775 |
| 5 | 0.5770 | 0.5554 | 0.5560 | 0.6130 | 0.5790 | 0.5998 | 0.5351 | **0.5273** |
| 7 | 0.6687 | 0.7076 | 0.7072 | 0.6796 | 0.6677 | 0.6743 | **0.6447** | 0.6506 |
| 9 | 0.8282 | 0.8412 | 0.8327 | 0.8243 | 0.8222 | 0.7953 | **0.7335** | 0.7503 |

## C  DETAILED PROOFS

*Proof of Lemma 3.1.* We can compute

$$\mathbb{P}(\boldsymbol{\delta}^i = \mathbf{0}) = 1 - \mathbb{P}\left(u_i \leq \Phi^{-1}\left(r_i(\gamma)\right)\right) = 1 - r_i(\gamma). \tag{C.1}$$

That is, with probability $1 - r_i(\gamma)$, $\boldsymbol{\delta}^i = 0$. Equivalently, $\boldsymbol{\delta}^i$ is distributed by a degenerated probability measure with Dirac density $D(\boldsymbol{\delta}^i)$ concentrated at 0. On the other hand, with probability $r_i(\gamma)$, $\boldsymbol{\delta}^i$ is distributed as $q_i'(\cdot|\mathbf{x}; \beta)$. Combining the two cases, it follows that $\boldsymbol{\delta}^i$ is distributed by a mixture of $q_i'(\cdot|\mathbf{x}; \beta)$ and $D(\boldsymbol{\delta}^i)$ with weights $r_i(\gamma)$ and $1 - r_i(\gamma)$ respectively. $\square$

*Proof of Lemma 3.2.* By the construction of $r_i(\gamma)$,

$$\mathbb{E}\left[s(\boldsymbol{\delta})\right] = \sum_{i=1}^{d} \mathbb{E}\left[\mathbb{I}\left(u_i \leq \Phi^{-1}\left(r_i(\gamma)\right)\right)\right] = \sum_{i=1}^{d} \mathbb{P}\left(u_i \leq \Phi^{-1}\left(r_i(\gamma)\right)\right)$$

$$= \sum_{i=1}^{d} r_i(\gamma) = \frac{\kappa}{\sqrt{d}} \cdot \frac{\sum_{i=1}^{d} \gamma_i^{1/2}}{\left(\sum_{i=1}^{d} \gamma_i\right)^{1/2}} \leq \kappa.$$

$\square$

**Table 7:** Average wQL on **Taxi** dataset under **deterministic** attack. Target time series $\mathcal{I} = \{1\}$ and attacked time stamp $H = \{\tau\}$. Smaller is better.

| Sparsity ($\kappa$) | no defense | data augmentation | randomized smoothing | mini-max defense |
|---|---|---|---|---|
| no attack | 1.2135±0.4050 | 1.2137±0.4091 | 1.2574±0.4281 | **1.0447**±0.3607 |
| 1 | 1.3152±0.4580 | 1.3455±0.4666 | 1.3455±0.4627 | **1.1222**±0.3960 |
| 3 | 1.6389±0.5810 | 1.6805±0.5982 | 1.6503±0.5756 | **1.3624**±0.4956 |
| 5 | 2.0317±0.7161 | 2.0625±0.7290 | 2.0123±0.7059 | **1.6830**±0.6206 |
| 7 | 2.3695±0.8064 | 2.3712±0.8028 | 2.3450±0.7978 | **1.9750**±0.7033 |
| 9 | 2.5605±0.8531 | 2.5525±0.8616 | 2.5422±0.8619 | **2.2374**±0.7785 |

**Table 8:** Average wQL on **Taxi** dataset under **probabilistic** attack. Target time series $\mathcal{I} = \{1\}$ and attacked time stamp $H = \{\tau\}$. Smaller is better.

| Sparsity ($\kappa$) | no defense | data augmentation | randomized smoothing | mini-max defense |
|---|---|---|---|---|
| no attack | 1.2118±0.4412 | 1.2526±0.4733 | 1.2241±0.4531 | **1.0481**±0.3840 |
| 1 | 1.4598±0.5315 | 1.3539±0.5199 | 1.3512±0.5100 | **1.1528**±0.4345 |
| 3 | 1.5659±0.6589 | 1.5446±0.6197 | 1.5567±0.5784 | **1.3940**±0.5472 |
| 5 | 1.9123±0.7513 | 1.7824±0.6962 | 1.8857±0.7441 | **1.6897**±0.6829 |
| 7 | 2.2915±0.8954 | 1.7340±0.7638 | 1.8370±0.7597 | **1.5865**±0.6191 |
| 9 | 2.4815±0.9286 | 2.1159±0.7515 | 2.2400±0.7860 | **1.4921**±0.5551 |

**Table 9:** Average wQL on **Wiki** dataset under **deterministic** attack. Target time series $\mathcal{I} = \{1\}$ and attacked time stamp $H = \{\tau\}$. Smaller is better.

| Sparsity ($\kappa$) | no defense | data augmentation | randomized smoothing | mini-max defense |
|---|---|---|---|---|
| no attack | 0.1645±0.0588 | 0.0868±0.0232 | **0.0796**±0.0272 | 0.2331±0.1186 |
| 1 | 0.2430±0.0889 | 0.0775±0.0171 | **0.0687**±0.0119 | 0.1683±0.1097 |
| 3 | 0.2771±0.0807 | 0.2225±0.1217 | 0.2260±0.1089 | **0.1466**±0.0976 |
| 5 | 0.4260±0.1127 | 0.3533±0.1602 | 0.3084±0.1365 | **0.1675**±0.0675 |
| 7 | 0.5173±0.1045 | 0.4290±0.1524 | 0.4112±0.1420 | **0.1973**±0.0632 |
| 9 | 0.6276±0.1178 | 0.4362±0.1360 | 0.4451±0.1461 | **0.2185**±0.1131 |

**Table 10:** Average wQL on **Wiki** dataset under **probabilistic** attack. Target time series $\mathcal{I} = \{1\}$ and attacked time stamp $H = \{\tau\}$. Smaller is better.

| Sparsity ($\kappa$) | no defense | data augmentation | randomized smoothing | mini-max defense |
|---|---|---|---|---|
| no attack | 0.1748±0.1144 | 0.0837±0.0432 | **0.0828**±0.0443 | 0.2376±0.1510 |
| 1 | 0.3255±0.2132 | **0.1647**±0.1126 | 0.1976±0.1274 | 0.1834±0.1409 |
| 3 | 0.4080±0.1724 | 0.3104±0.1550 | **0.2255**±0.1322 | 0.2549±0.1530 |
| 5 | 0.5336±0.2318 | 0.2759±0.1368 | 0.1714±0.1348 | **0.1299**±0.0852 |
| 7 | 0.6547±0.2940 | 0.3940±0.1849 | 0.2656±0.1708 | **0.2569**±0.1605 |
| 9 | 0.8463±0.2715 | 0.5140±0.2195 | **0.2745**±0.1513 | 0.2909±0.1918 |

*Proof of Theorem 4.1.* Denote $p_\sigma(\cdot)$ as the density of $\mathbb{N}(0, \sigma^2 \mathbf{I}_d)$ and $p(\cdot)$ as the density of $\mathbb{N}(0, \mathbf{I}_d)$. Let $F_{\mathbf{x}}(\mathbf{r}) \triangleq \mathbb{P}(z(\mathbf{x}) \preceq \mathbf{r})$. Consider

$$
\begin{aligned}
\sup_{\mathbf{r} \in \mathbb{R}^d} \left| G(\mathbf{r}) - G_{\boldsymbol{\delta}}(\mathbf{r}) \right| &= \sup_{\mathbf{r} \in \mathbb{R}^d} \left| \int_{\boldsymbol{\epsilon} \in \mathbb{R}^{d \times T}} \left( F_{\mathbf{x}+\boldsymbol{\epsilon}}(\mathbf{r}) - F_{\mathbf{x}+\boldsymbol{\delta}+\boldsymbol{\epsilon}}(\mathbf{r}) \right) p_\sigma(\boldsymbol{\epsilon}) \, d\boldsymbol{\epsilon} \right| \\
&= \sup_{\mathbf{r} \in \mathbb{R}^d} \left| \int_{\boldsymbol{\epsilon} \in \mathbb{R}^{d \times T}} F_{\boldsymbol{\epsilon}}(\mathbf{r}) \left( p_\sigma(\boldsymbol{\epsilon} - \mathbf{x}) - p_\sigma(\boldsymbol{\epsilon} - \mathbf{x} - \boldsymbol{\delta}) \right) d\boldsymbol{\epsilon} \right| \\
&= \sup_{\mathbf{r} \in \mathbb{R}^d} \left| \int_{\boldsymbol{\epsilon} \in \mathbb{R}^{d \times T}} \int_0^1 F_{\boldsymbol{\epsilon}}(\mathbf{r}) \nabla p_\sigma(\boldsymbol{\epsilon} - \mathbf{x} - t\boldsymbol{\delta}) \boldsymbol{\delta} \, dt \, d\boldsymbol{\epsilon} \right| \\
&= \sup_{\mathbf{r} \in \mathbb{R}^d} \left| \int_0^1 \int_{\boldsymbol{\epsilon} \in \mathbb{R}^{d \times T}} F_{\boldsymbol{\epsilon}}(\mathbf{r}) \left( \boldsymbol{\delta} \cdot \frac{\boldsymbol{\epsilon} - \mathbf{x} - t\boldsymbol{\delta}}{\sigma^2} \right) p_\sigma(\boldsymbol{\epsilon} - \mathbf{x} - t\boldsymbol{\delta}) \, d\boldsymbol{\epsilon} \, dt \right| \\
&= \frac{1}{\sigma} \sup_{\mathbf{r} \in \mathbb{R}^d} \left| \int_0^1 \int_{\boldsymbol{\epsilon} \in \mathbb{R}^{d \times T}} F_{\mathbf{x}+t\boldsymbol{\delta}+\boldsymbol{\epsilon}}(\mathbf{r}) \left( \boldsymbol{\delta} \cdot \boldsymbol{\epsilon} \right) p(\boldsymbol{\epsilon}) \, d\boldsymbol{\epsilon} \, dt \right| \\
&\leq \frac{1}{\sigma} \int_{\boldsymbol{\epsilon} \in \mathbb{R}^{d \times T}} \left| \boldsymbol{\delta} \cdot \boldsymbol{\epsilon} \right| p(\boldsymbol{\epsilon}) \, d\boldsymbol{\epsilon} \\
&\leq \frac{\|\boldsymbol{\delta}\|_2}{\sigma} \left( \mathbb{E}_{\boldsymbol{\epsilon} \sim \mathbb{N}(0, I_d)} \|\boldsymbol{\epsilon}\|_2^2 \right)^{\frac{1}{2}} = \frac{\sqrt{d}}{\sigma} \|\boldsymbol{\delta}\|_2,
\end{aligned}
$$

**Table 11:** Average wQL on **Traffic** dataset under **deterministic** attack. Target time series $I = \{1\}$ and attacked time stamp $H = \{1\}$. Smaller is better.

| Sparsity ($\kappa$) | no defense | data augmentation | randomized smoothing | mini-max defense |
|---|---|---|---|---|
| no attack | 0.2731±0.1606 | 0.2831±0.1626 | 0.2686±0.1505 | **0.2452**±0.1367 |
| 1 | 0.2902±0.1523 | 0.2707±0.1374 | 0.2834±0.1287 | **0.2530**±0.1425 |
| 3 | 0.3779±0.1881 | 0.3409±0.1472 | 0.3304±0.1562 | **0.2947**±0.1543 |
| 5 | 0.4396±0.2042 | 0.3813±0.1726 | 0.3926±0.1573 | **0.3401**±0.1646 |
| 7 | 0.5358±0.2369 | 0.5009±0.2391 | 0.4364±0.2501 | **0.3562**±0.1737 |
| 9 | 0.6479±0.2875 | 0.4881±0.2546 | 0.5165±0.2532 | **0.3822**±0.1775 |

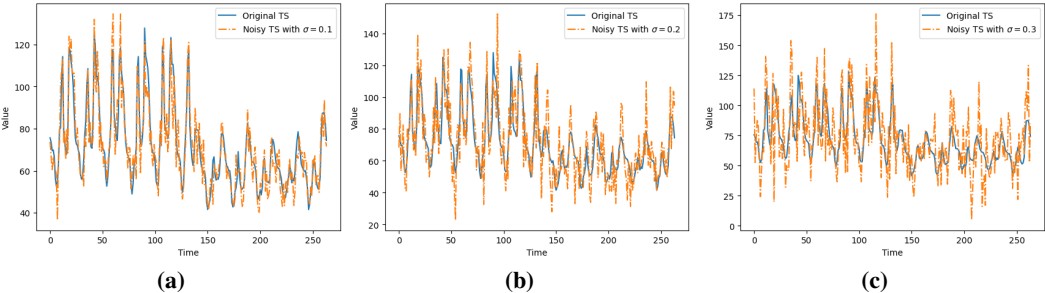

(a)          (b)          (c)

**Figure 4:** Plots of original time series $\mathbf{x}$ (blue) vs noisy time series $\tilde{\mathbf{x}}$ (orange) under different noise level $\sigma = 0.1, 0.2, 0.3$, where $\tilde{\mathbf{x}} = \mathbf{x}(1 + \sigma * \mathcal{N}(0, 1))$.

which completes the proof.      □

## D   Non-transferrability of attacks between univariate and multivariate forecasters

We study the transferrability from univariate attack to multivariate attack. To be specific, if an attack is generated on the same subset (excluding target time series) of time series using a univariate model and then fed into a multivariate model, can it indirectly harm the prediction of target time series. Next, we report the experiment results of univariate attack and multivariate attack.

**Table 12:** Transfer the attack from DeepAR to DeepVAR. Target items $\mathcal{I} = \{1\}$ and time horizon to attack $H = \{\tau\}$. Clean DeepAR and DeepVAR models are used. Averaged wQL is reported.

| No attack | Univariate attack | Multivariate attack |
|---|---|---|
| 0.288 | 0.322 | 0.390 |

## E   Time series under different noise level

For better illustration, we also plot the history of the noisy time series used in Appendix D. That is, the electricity usage of the fifth user over $T = 1, \ldots, 264$.

## F   Attack and defense on deterministic model

In this section, we conduct parallel experiments to examine the attack and defense effect on deterministic (non-probabilistic) forecasting models. We implement a SOTA Informer model (Zhou et al., 2021) on Traffic dataset. We set the hyper-parameters exactly the same as those in Table 1 for fair comparison. As Informer model is non-probabilistic, we only report MSE loss below in Table 13.

**Table 13:** Average MSE on **Traffic** dataset under **deterministic** attack on **Informer** model with $\tau = 24$. Target time series $\mathcal{I} = \{1, 5\}$ and attacked time stamp $H = \{\tau - 1, \tau\}$. Smaller is better.

| Sparsity ($\kappa$) | no defense | data augmentation | randomized smoothing | mini-max defense |
|---|---|---|---|---|
| no attack | $0.0552 \pm 0.0033$ | $0.0513 \pm 0.0029$ | $\mathbf{0.0492 \pm 0.0004}$ | $0.0644 \pm 0.0019$ |
| 1 | $0.0554 \pm 0.0030$ | $0.0515 \pm 0.0032$ | $\mathbf{0.0492 \pm 0.0003}$ | $0.0642 \pm 0.0021$ |
| 3 | $0.1058 \pm 0.0073$ | $0.0999 \pm 0.0060$ | $\mathbf{0.0959 \pm 0.0029}$ | $0.1339 \pm 0.0051$ |
| 5 | $0.3390 \pm 0.0134$ | $0.2617 \pm 0.0102$ | $\mathbf{0.2551 \pm 0.0061}$ | $0.4568 \pm 0.0151$ |
| 7 | $0.7279 \pm 0.0251$ | $0.5139 \pm 0.0157$ | $\mathbf{0.4997 \pm 0.0087}$ | $1.1257 \pm 0.0263$ |
| 9 | $1.3144 \pm 0.0454$ | $0.8285 \pm 0.0273$ | $\mathbf{0.8081 \pm 0.0167}$ | $2.1817 \pm 0.0490$ |

## G  LONGER PREDICTION LENGTH

In this section, we evaluate our attack and defense effect on Informer model with longer prediction length. We set the prediction length $\tau = 168$ and the other hyper-parameters are exactly the same as those in Table 1 and Table 13. The experiments results are reported in Table 14.

**Table 14:** Average MSE on **Traffic** dataset under **deterministic** attack on **Informer** model with $\tau = 168$. Target time series $\mathcal{I} = \{1, 5\}$ and attacked time stamp $H = \{\tau - 1, \tau\}$. Smaller is better.

| Sparsity ($\kappa$) | no defense | data augmentation | randomized smoothing | mini-max defense |
|---|---|---|---|---|
| no attack | $0.3593 \pm 0.0132$ | $0.3770 \pm 0.0344$ | $\mathbf{0.3425 \pm 0.0042}$ | $0.3941 \pm 0.0052$ |
| 1 | $0.3565 \pm 0.0125$ | $0.3721 \pm 0.0368$ | $\mathbf{0.3436 \pm 0.0035}$ | $0.3947 \pm 0.0057$ |
| 3 | $0.4556 \pm 0.0147$ | $0.4377 \pm 0.0386$ | $\mathbf{0.3998 \pm 0.0050}$ | $0.4516 \pm 0.0075$ |
| 5 | $0.6057 \pm 0.0195$ | $0.5979 \pm 0.0449$ | $\mathbf{0.5522 \pm 0.0081}$ | $0.6355 \pm 0.0112$ |
| 7 | $0.8656 \pm 0.0262$ | $0.8871 \pm 0.0764$ | $\mathbf{0.8242 \pm 0.0178}$ | $0.9009 \pm 0.0157$ |
| 9 | $1.2142 \pm 0.0283$ | $1.2944 \pm 0.0992$ | $1.2340 \pm 0.0204$ | $\mathbf{1.1778 \pm 0.0168}$ |

