# OpenReview forum: "Robust Multivariate Time-Series Forecasting: Adversarial Attacks and Defense Mechanisms"
_ICLR.cc/2023/Conference — ICLR 2023 poster_

### Official Review · Reviewer_K2Yj · 2022-10-23

**Confidence:** 3
**Correctness:** 4
**Technical Novelty And Significance:** 3
**Empirical Novelty And Significance:** 3
**Recommendation:** 6

**Clarity, Quality, Novelty And Reproducibility:**

The paper seems to demonstrate original ideas and is clearly and understandably written.

**Strength And Weaknesses:**

The paper is well-written and motivated, covering many nuances of the problem at hand.

**Summary Of The Paper:**

The paper adapts adversarial attack strategies to multivariate probabilistic forecasting models, focusing on attacks that might be difficult to detect. They also adapt defense mechanisms to adversarial attacks to the time-series forecasting setup and demonstrate how multivariate problems differ from univariate ones.

**Summary Of The Review:**

The paper answers a natural research question on defending against adversarial attacks in time series forecasting problems. This is a natural and well-motivated problem that is quite clearly answered and hence a non-trivial contribution to the literature.

---

> ### Author Response · Authors · 2022-11-15
> **Response to Reviewer K2Yj**
>
> Thanks for your comments. We especially thank you for pointing out that the paper demonstrates original ideas and recognizing our work as a non-trivial contribution.
> Please don’t hesitate to let us know if you have any additional comments or questions in the rebuttal period.

---

### Official Review · Reviewer_oCqG · 2022-10-24

**Confidence:** 4
**Correctness:** 4
**Technical Novelty And Significance:** 3
**Empirical Novelty And Significance:** 3
**Recommendation:** 6

**Clarity, Quality, Novelty And Reproducibility:**

To elaborate on my concerns above:
- the threat model described in 3.1 seems directly taken from Dang-Nhu et al. (which you are already citing). In particular you are reusing the notation $\chi$ and $t$ which was introduced by this paper. This is fine but it would make sens to properly attribute this.
- as far as I understand, the "mini-max" defence falls into the category of standard adversarial training? I'm concerned that this section is only citing GAN papers which are distantly related. Could you please comment of the novelty of this method and properly discuss related work?
-  My memories of randomised smoothing are somewhat distant but as far as I understand, theorem 4.1. is a rather standard and application-independent result? Are you able to explain what differences in the proof allow you to obtain a result valid for any $\delta$ and to apply the result to a multivariate setting?


I would also like to clarify some aspects of the contribution:
- In the deterministic attack (3.2), since the sparsity constraint can be solved analytically, would it make sense to apply to use projected gradient descent also for this and project at each optimisation step rather than just at the end?
- In your description of the probabilistic attack (3.3.), you are immediately factoring the distribution (last line of page 4). Not every distribution can be factored in this way. If you choose to make some kind of independence assumption in the modelling, you need to state it clearly.
- The math of the probabilistic attack (3.3) appear sound to me, but I find this section to be a bit terse in explanations
    - In the final step of algorithm 4, you are simply drawing $\delta$ from the distribution. It is proven that *in expectation* this $\delta$ will have the desired level of sparsity, but this is a probabilistic guarantee. Do you actually check that the drawn perturbation is sparse?
    - I feel like I'm lacking some context about this probabilistic relaxation of the sparsity constraint. Is there a body of related literature? This section doesn't mention any kind of related work
     - Algorithm 4 doesn't explain how you compute or approximate the outer expectation in equation (3.5). Does this build on the reparametrization in theorem 3.1? Is this a standard approach to reparametrizing discrete variables in a model?
- In 5.2, you refer to increased $\kappa$ as "increased sparsity". Intuitively I would see this as "decreased sparsity"

**Strength And Weaknesses:**

I find the main claim to be well supported as the paper covers all aspects of adversarial attacks on multivariate probabilistic forecasting. I am mostly satisfied with the mathematical treatment and the experiments appear sound.

I have ambivalent feelings about the description of related work. The paper clearly differentiates from prior attacks on probabilistic time series forecasting by its multivariate setting. However, the presented attack and defense algorithms seem to be somewhat incremental on prior work, which creates a novelty concern.



**Summary Of The Paper:**

The submission studies adversarial attacks and defences in the context of multivariate probabilistic forecasting of time series. The threat model assumes that the attacker can perturb past values and that the sets of target series and perturbed series are distinct, with an additional sparsity contraint on the set of perturbed series.

**Summary Of The Review:**

I find the topic of this paper very interesting and the main claim to be well supported. However I have many questions and concerns about the clarity and novelty of the different algorithms described in the paper. I think that some additional explanations and improvements are needed to pass the acceptance bar.

---

> ### Author Response · Authors · 2022-11-15
> **Response to Reviewer oCqG**
>
> Thanks for your fruitful comments. We would like to address your concerns and please see our clarifications below.
>
> # 1 Novelty
> We would like to highlight our novelty when it comes to multivariate settings. First, we propose a novel and interesting problem **sparse indirect** attack, which doesn’t make sense in the univariate setting as there is no correlation between time series. Second, the sparsity level is uniquely brought by multivariate setting and doesn’t exist in univariate cases. Third, we used novel and different technique to prove the robustness certification of randomized smoothing. Note that the original proof of univariate RS as in [Yoon et al, 2022] is not directly applicable, as we remarked at the top of Page 7. To be specific, their proof of Lemma 2 in Appendix A.3 of Page 16 in [Yoon et al, 2022] is not available for multivariate case. This is because in the 3rd equation of their Page 16, the second equality used the fact that for univariate distribution function, $1 - P(f(z)\leq r) =P(f(z) > r)$. This doesn’t hold for multivariate distribution function. We also would like to refer the reviewer to general response for the novelty of this work.
>
> # 2 Reference
> We thank the reviewer for the suggestion regarding the notation in Sec. 3.1. We already reflected the suggestion in our revision.
>
> # 3 Re-parameterization
> We did use re-parametrization to compute the gradient of the outer expectation in Eqn (3.5). We’ve already made this clear in the revision at the end of Sec 3. This trick is standard in computing gradient of expectation and is also the default approach in PyTorch.
>
> # 4 Sparse layer
> We did check the sparsity of the drawn perturbation. We show some statistics regarding the sparsity of the perturbation. For example, with sparsity $\kappa=5$, the $s(\delta) = 4.44\pm 1.48$ based on 1000 repetitions. This is in accordance with our theoretical results, where we showed that $E[s(\delta)]\leq\kappa$ in Theorem 3.2.
>
> The differentiable probabilistic sparse layer we devised is novel. A somewhat related work is sparse max [1], which is a generalization of soft-max to convert output to sparse logits. However, different from [1], (i). our sparse layer has model parameters and are trainable. (ii). The input and output doesn’t have to have the same dimension in our sparse layer, as opposed to sparse max. (iii). The output of our sparse layer doesn’t have to be logits, meaning that it can take value outside $[0,1]$, while sparse max only outputs logits between 0 and 1 and sum up to 1.
>
> # 5 Discussion of randomized smoothing
> We explained why the proof in [Yoon et al, 2022] can’t be directly applicable to multivariate case in # 1. We would like to further explain that we consider the function $L_\infty$-norm here, while in [Yoon et al, 2022], they consider 1-Wasserstein distance between distributions. We found that $L_\infty$-norm is more convenient to analyze, can be generalized to multivariate cases, and can yield a conclusion that holds for any $\delta$. This is the reason why we mainly consider $L_\infty$-norm here.
>
> # 6 Discussion of mini-max defense
> We would like to highlight the novelty of mini-max defense first. The mini-max defense uses the sparse layer that we proposed in Sec. 3.3. Since the sparse layer is differentiable, it allows end-to-end optimization. This is why it can be used in defense. Since we are the first to propose a differentiable probabilistic sparse layer, to the best of our knowledge, we couldn’t find any more closely related reference to properly cite.
>
> # 7 Independence
> We thank the reviewer for the suggestion. We do need independence assumption between the rows of $\delta$, which was already reflected it in our revision.
>
> # 8 Increasing sparsity
> We thank the reviewer for the suggestion on the terminology. To avoid confusion, we have replaced the smaller/larger sparsity level by smaller/larger $\kappa$ in Sec 5 of our revision.
>
> # 9 Another method in det. attack
> We thank the reviewer for proposing another method to design deterministic attack and we agree this method might yield more effective attacks. We will carefully consider and include this result if it gives noticeably better attacks. Nevertheless, even if the projection is only taken at the last step, the attack is still powerful as it increases the wQL loss by up to 190% from Table 1. We kindly refer the reviewer to Sec 5 for more detailed discussion.
>
> We believe we have addressed all your concerns. Please don’t hesitate to let us know if you have any additional comments or questions in the rebuttal period.
>
> Reference:
> [1] Martins, Andre, and Ramon Astudillo. "From softmax to sparsemax: A sparse model of attention and multi-label classification." In International conference on machine learning, pp. 1614-1623. PMLR, 2016.

---

> > ### Comment · Reviewer_oCqG · 2022-11-15
> > **Answer to authors**
> >
> > I thank the authors for their detailed comments which clarified some misunderstandings on my side. Based on these additional explanations, the novelty of the contribution appears more clearly. Some technical aspects are also easier to understand. Perhaps the main point that remains unclear to me is the relation between the mini-max defense and standard adversarial training methods which also have a min-max loss.
> >
> > To reflect on these improvements, **I increased my novelty ratings to 3 and I changed to an acceptance score of 6**.

---

### Official Review · Reviewer_hzNw · 2022-10-25

**Confidence:** 3
**Correctness:** 3
**Technical Novelty And Significance:** 2
**Empirical Novelty And Significance:** 3
**Recommendation:** 6

**Clarity, Quality, Novelty And Reproducibility:**

The paper is well-written. The solution is sort of similar to existing adversarial attacks bu only a few papers that are addressed the adversarial attack issue in time series forecasting. I did not find the source code attached to the manuscript.

**Strength And Weaknesses:**

Strength
1. The paper touches on an interesting paper on time series forecasting
2. The experiment is solid

Weakness
1. The high-level idea of the method is very similar to the classical techniques used in other closed research fields such as the time series of classification as [1]
2. Proposed method only evaluated on DeepAR and DeepVAR
3. Lacking of graphical examples of how time series in different noise-level looks like.

Detailed Comments that needs to be addressed by authors: See the summary of review.

**Summary Of The Paper:**

In this paper, authors introduced an optimization-based adversarial attack to in-direct attack the forecasting performance. In addition, the authors proposed an protection approach to protect the model from the proposed attack.

**Summary Of The Review:**

1. The proposed approach was only tested with DeepAR and DeepVAR forecasting models. First of all, why only tested in DeepAR-based models, which is not the optimal solution in general? For example, is the proposed work also can protect the Deep State Space Model from attacking? How about other non-probabilistic forecasting models? For example, NBeats or Informer.

2. The technique used here is very standard. While only limited work focuses on time series forecasting, a lot of work has been done in time series classification [1]. The high-level idea of formalizing the objective equation is very similar. The authors should mention what is the benefit of the novel loss function.

3. In the experiment, the model is evaluated on WQLoss. Why do we only use WQLoss? Intuitively, CRPS, RMSE, and MAE are all valid evaluation criteria. Besides, from Fig. 1, I personally do not see the attack as a success because it does not change a lot of the forecasting. You need to show what the forecasting with confidence interval looks like to justify your claim.

4. If using i.i.d based forecasting, what will happen for the in-direct attack?

* Thank you for the discussion. I believe the authors successfully resolve most of my concerns. I increase my score to 6.

5. The forecasting length is short compared with the long-term forecasting model. What is the performance of different lengths? Compared the performance of different lengths may be needed.

6. Mini-max Defense is specially designed for the proposed attack, but if the attack is not that aggressive, what somethings could happen?

7. Visually speaking, how are the noisy-added time series look under different noise-level?

[1] Zhang T, Liu S, Wang Y, Fardad M. Generation of low distortion adversarial attacks via convex programming. In2019 IEEE International Conference on Data Mining (ICDM) 2019 Nov 8 (pp. 1486-1491). IEEE.

---

> ### Author Response · Authors · 2022-11-15
> **Response to Reviewer hzNw (1/2)**
>
> Thanks for your fruitful comments and discussion. We would like to address your concerns and please see our clarifications below.
>
> # 1 Novelty in differentiating from classification settings
> We thank the reviewer for pointing out the reference [1]. Here, we want to re-emphasize that neither the setting of time series and techniques can directly be adopted from image classification setting in [1] (which is also well-emphasized in the previous works [Dang-Nhu et al, 2020], [Yoon et al. 2022]). Unlike classification setting  with their **independent** samples assumed, **time series** datasets (electricity, traffic, taxi, wiki, etc.) have both temporal correlations and item-wise correlations and thus **probabilistic multivariate, multi-horizon** forecasters have been separately designed. This opens up new challenges in designing attack and defense mechanism accordingly.
>
> To be specific, we also consider **sparse indirect** attack scenario, which has not been considered by general classification setting including [1]. In fact, to the best of our knowledge, we are the first to propose such attacks on multivariate time series forecasting models. Note that this attack makes no sense in classification and univariate time series setting as there is no clear notion of items, time horizon, and their correlations.
>
> Moreover, the effective attacks should be designed differently due to its temporal nature, especially under auto-regressive models. [1] would like to design attacks that make the classifier yield a wrong single predicted label. In this case, the predicted label is never fed into the classifier again so an attack can be designed to aggressively change the predicted label. However, in time series setting, one-step prediction will further be fed into forecaster to make a next step prediction. With this significant difference, the most effective time series attacks do not  only greedily change the next step prediction as it has to consider changing multi-step predictions. Also, there are other significant differences between forecasting and classification, such as correlation over multiple time series, and probabilistic predictions. We also kindly refer to our main reference of univariate forecasting setting [Yoon et al., 2022] for parallel explanations.
>
> Finally, we also propose a novel differentiable probabilistic attack to achieve sparse indirect attack using a sparse layer, which is never investigated by prior work including [1]. We adopt randomized smoothing in multivariate case, where we use different and novel techniques to prove a more general robustness guarantee which holds for any $\delta$ (vs previous work [Yoon et al, 2022] whose certification only holds for $\delta\to0$). We propose a mini-max defense that beats baseline defense mechanisms.
>
> We also would like to refer the reviewer to general response for the novelty of this work.
>
> # 2 More metrics and confidence interval
> We did report more metrics like Weighted-MSE (WMSE) and Weighted MAE (WAPE) loss (both defined in appendix B.4) in Table 4 & 5 in appendix Page 14 with confidence intervals. But for better presentation, they are not shown in the main text. Besides, we would like to clarify that in figure 1, the attack is only designed for the last time index ($T=288$), not for all time indices. At $T=288$,  the ground truth is 93.2, clean prediction is $110.2\pm4.68$ with 95% CI, prediction under attack is $129.3\pm5.82$ with 95% CI, which is outside the CI of clean prediction. Moreover,  **our attack has substantially increased the MAE loss by 112%**. Recall that this is only the attack with sparsity 1, the weakest setting. With increased attacking sparsity, the effect will be even more noticeable.
>
> # 3 Time series under different noise level
> Thanks for suggesting to add more graphical presentations. We have plotted the requested graph in Appendix E of our revision for better visualization of how the noisy time series look under different noise levels $\sigma$. These noisy time series data are used in data augmentation and randomized smoothing.
>
> # 4 Mini-max on non-aggressive attack
> The aggressiveness of an attack can be reflected by its sparsity level. The attack with smaller sparsity $\kappa$ is considered to be less aggressive. Our mini-max defense also performs well (although not optimal) on less aggressive attacks. For example, from Table 2, mini-max is the best for sparsity level >=3. Although it’s not better than RS for $\kappa=0,1$, it’s still better than the model without defense.

---

> > ### Author Response · Authors · 2022-11-15
> > **Response to Reviewer hzNw (2/2)**
> >
> > # 5 Evaluation on different types of model
> > We thank the reviewer for encouraging us to implement our methodology on more types of models and asking for the performance on **deterministic** models like Informer, despite that our focus is **probabilistic** models. As mentioned in the introduction, note that uncertainty quantification is commonly required for downstream tasks and thus often of necessity. That being said, among probabilistic forecasters, DeepAR are popular SOTA models and have been extensively studied in the previous literature (such as [Dang-Nhu et al, 2020], [Yoon et al. 2022]). Therefore we consider DeepVAR, a generalization of DeepAR,  as a backbone SOTA probabilistic model for multivariate time series.
> >
> > Even so, here we additionally included the results of Informer in Appendix F in our revision. Due to limited rebuttal time, we only evaluate our attack and defense on Informer model on Traffic datasets. For fair comparison, the experiment setting is exactly the same as that in our Table 2 except the metrics. Since Informer is a non-probabilistic model, we cannot compute wQL. Instead, we report MSE loss in Table 12 in Appendix F. From this table, we conclude that (i). Our deterministic attack still works and can increase MSE loss by up to 2200%! (ii). Our defense (RS) still works for Informer model and outperforms baseline defense (no defense & data augmentation). We will report more detailed experiments on Informer in revision for readers.
> >
> > # 6 Prediction length
> > We thank the reviewer for encouraging us to try longer prediction length. We have included the additional results in our Table 13 in Appendix G. In the additional experiments, we only trained an informer model with prediction length 168 on traffic dataset due to limited rebuttal time. Whenever possible, we keep the other hyper-parameters the same as those in our Table 2. To conclude, even with longer prediction length, our attack algorithm is still effective since it can increase the MSE loss by up to 238%. Moreover, our defense (RS) also outperforms baseline defense (no defense and data augmentation).
> >
> > # 7 I.I.D based forecaster
> > We thank the reviewer for proposing this question. As there are multiple dimensions of multivariate time series, we would like to kindly ask the reviewer to clarify what do you mean by i.i.d. based forecaster. If i.i.d. here means each time series is i.i.d., then the problem can be converted to univariate case, which has been discussed in [Yoon et al., 2022]. If the i.i.d. means non-autoregressive forecaster, then the forecaster will be a seq-2-seq model. In this case, one needs to find a probabilistic, multivariate, seq-2-seq forecasting model first, so we think it’s still an open question in a probabilistic setting.
> >
> > We believe we have addressed all your concerns. Please don’t hesitate to let us know if you have any additional comments or questions in the rebuttal period.

---

> > > ### Comment · Reviewer_hzNw · 2022-11-17
> > > **Response to Author**
> > >
> > > Thank you very much for the author's comments. It helps me understand the paper much more clearly. Overall, I am leaning to change the score, but I still have several questions that the authors need to answer:
> > >
> > > Here are the questions:
> > >
> > > 1. EQ 3.1 does not seem like taking the characteristic of time series forecasting mentioned by authors (e.g. seq2seq prediction, high correlation) into consideration. After reading Yoon 2022 paper, I believe the reason is because of lacking explanation. For example, what is t_adv and \Chi{z} is only explained at a very high level. I personally think adding examples of  t_adv and \Chi{z} used to design attack is crucial to increase the clarity of the draft.
> > >
> > > 2. Which alpha you used for wQL(alpha)?
> > >
> > > 3. In most experiments, the model only attacks and protect countable time stamps (mostly focuses on the farthest future timestamp)? Can you explain any reasons why the model only targets this time stamp?
> > >
> > > 4. Any reason why authors do not use the Deep State Space Model?
> > >
> > > 5. The t_{adv} is chosen from two values 0.5 and 2. What does this mean? Do you use both t_adv during the attack?
> > >
> > > 6. Why do all the data used have very high dimensions? Any reason for choosing so high dimension time series?
> > >
> > > 7. A simple defense is just using univariate time series to do the forecasting (e.g. every dimension just forecasts separately). Any comments on this case?
> > >
> > > 8. Lastly, since the time series authors used is the high dimension, given a time stamp t. there potentially exist some dimensional that may be i.i.d toward the target dimension. (e.g. in a 3-dimensional time series [t1,t2,t3], the third dimension time series t3 is recorded independent of dimension 1 time series t1, but we do not know it beforehand). This scenario often exists in a lot of traffic time series with large sensor networks (since there are some sensors geographically far away from each other, we almost can assume the signal is just recorded without any correlation). So if this type of t_i existed in the multivariate time series, will the proposed attack method fail to indirectly attack the forecasting result?

---

> > > > ### Author Response · Authors · 2022-11-19
> > > > **Response to Reviewer hzNw (1/2)**
> > > >
> > > > Thanks for the reply and comments. We would like to address your concerns as follows.
> > > >
> > > >
> > > > 1. We thank the reviewer for this suggestion to further improve our manuscript. We have reflected the suggestions by adding a concrete example for $t_\text{adv}$ and $\chi(z)$ in the 1st paragraph of Sec 3.1 in our revision.
> > > >
> > > > 2. As mentioned in the last line of Appendix B.3, we select $\alpha$ from $[0.1,0.2,\dots, 0.9]$ and compute the average wQL on these 9 values.
> > > >
> > > > 3. We did study the effect of different hyper-parameters, like varying target items, attacked time horizon, sparsity level, and noise level. The selection of the attacked time horizon is rather arbitrary and our methods can be applied to any (or multiple) time horizons. We have included additional experiments in Table 11 of Appendix B.4 in revision. In this table, we conduct a deterministic attack on time stamp $T=1$, i.e. the first time stamp, and we can still come to the same conclusion:
> > > > * The attack is effective as it can increase the wQL by up to 137%.
> > > > * Our defense (mini-max) achieves the best wQL at all sparsity levels.
> > > >
> > > > However, for defense methods, we are not protecting a specific time stamp. Recall that randomized smoothing is a post-processing technique to enhance overall model robustness. Also recall that our mini-max defense doesn’t take any attacker’s information (e.g. target time series, attacked time stamps) as its hyper-parameter, as it doesn’t have access to attacker’s information. Although these defense mechanisms view the attack as a black-box one, they still work well and beat baseline method, see our Table1,2,7,8,9,10, etc.
> > > >
> > > > 4. We kindly want to note that DeepVAR, which we used as a backbone of the experiments, can also be seen as a state space model with RNN hidden (state) variables. It is not clear what specific deep state space model is being referred to by the reviewer, and thus we could not judge the necessity of repeating all with another backbone model. Still, to the best of our knowledge, we found out one popular SOTA that is explicitly named after state space with capability of *probabilistic* modeling is [4], which was only compared with global (univariate) modeling class like DeepAR, but *not multivariate* modeling like DeepVAR in the experiments, and thus we decided to exclude this model (even though they might be capable of multivariate probabilistic modeling in theory). Another practical reason is that DeepVAR is recently re-written in PyTorch, which we most prefer, but Deep State Space model like [4] is written in MXNET. Instead, as we responded in our first rebuttal, we focused on including experiments of point (deterministic) forecaster class, Informer, beyond probabilistic forecasters, requested by the reviewer to respond at our best given the limited rebuttal period.
> > > >
> > > > 5. Yes, as we stated in the last paragraph of Page 13 that
> > > > *We report the largest error produced by these choices of constants.*
> > > > This is also the convention to generate un-targeted adversarial attack in [Yoon et al, 2022] and we have made it clearer in the revision.
> > > >
> > > > 6. The dimension of the dataset could be arbitrary and our method doesn’t have any requirement on how high the dimension should be (but at least 2-D due to multivariate). For the datasets, we simply select a few benchmarking time series datasets that are used by other popular literatures ([1], [2], [3], etc.), instead of intentionally selecting those with high dimensions.
> > > >
> > > > 7. We thank the reviewer for providing such suggestions. However, even before we develop attack and defense strategies, we have to make sure that the forecasting model is a good one. (Otherwise, as an extreme case, imagine that a trivial forecaster only predicts 0 all the time, then no attack will be effective.) However, if multiple time series are correlated, univariate forecasters can’t do a good job (see [1]), so there is no point to develop attacks and defenses for these weak models. In other words, if we want to apply the reviewer’s suggestion, we need to check the correlation between the TS to make sure univariate models can give a good prediction. However, our proposal is a more general method to enhance robustness of multivariate forecaster, no matter whether they are correlated or uncorrelated, or only a subset are correlated. But we do agree that in the case where time series are not strongly correlated, univariate model will be a reasonable choice and developing defense on univariate model (as discussed in [3]) will work well.

---

> > > > > ### Author Response · Authors · 2022-11-19
> > > > > **Response to Reviewer hzNw (2/2)**
> > > > >
> > > > >
> > > > > 8. It depends. First, we would like to clarify the underlying assumption of our work is that we consider correlated multivariate time series forecasting (otherwise it has been discussed in [3]).
> > > > > Let’s still consider the reviewer's example where $[t_1,t_22,t_3]$ is given and $t_3$ is independent of $t_1$. In this case, if $t_1$ and $t_2$ are not independent and we wish to indirectly attack $t_1$, we will end up selecting $t_2$ to attack (for sparsity level = 1). However if $t_1$ and $t_2$ are also independent (i.e. $t_1$ is independent of the other two), our method might fail because this independence is against our underlying assumption. For the application of traffic sensors as the reviewer described, suppose there is a traffic sensor far away from all the others, our method might fail as this sensor can be considered independent from the set of all the other sensors. If this is not the case, our method still works and will end up selecting a few sensors near this sensor to attack.
> > > > >
> > > > > We believe we have addressed all your concerns. Please don’t hesitate to let us know if you have additional comments.
> > > > >
> > > > > References:
> > > > >
> > > > > [1]. Salinas, David, et al. "High-dimensional multivariate forecasting with low-rank gaussian copula processes." Advances in neural information processing systems 32 (2019).
> > > > >
> > > > > [2]. Dang-Nhu, Raphaël, et al. "Adversarial attacks on probabilistic autoregressive forecasting models." International Conference on Machine Learning. PMLR, 2020.
> > > > >
> > > > > [3]. TaeHo Yoon, Youngsuk Park, Ernest K Ryu, and Yuyang Wang. Robust probabilistic time series forecasting. In International Conference on Artificial Intelligence and Statistics, pp. 1336–1358. PMLR, 2022.
> > > > >
> > > > > [4] Deep state space models for time series forecasting, Syama Sundar Rangapuram, Matthias W. Seeger, Jan Gasthaus, Lorenzo Stella, Yuyang Wang, Tim Januschowski, NeurIPS 2018

---

### Official Review · Reviewer_qQby · 2022-10-31

**Confidence:** 5
**Correctness:** 4
**Technical Novelty And Significance:** 4
**Empirical Novelty And Significance:** 4
**Recommendation:** 8

**Clarity, Quality, Novelty And Reproducibility:**

Clarity and Quality: The main claims and experiments in this paper are well-described. The authors should consider providing the runtime of running attacks and defenses. Further, they should also study the attack performance when time series other than {1} are attacked as well as in the case when multiple target series can be attacked. What is the value of the threshold for the perturbation at each coordinate in the experiments? It is also not clear if the randomized defense designed here is a certified one or only provides empirical protection. If its a certified defense, then what guarantees does it give?

Novelty: The high-level idea behind the deterministic attack presented here has been explored for images in https://arxiv.org/pdf/1909.05040v1.pdf. The algorithm for the differentiable probabilistic attack is new.  The randomized smoothing defense is a straightforward adaptation of RS for other domains while the formulation of the mini-max defense is novel.

Reproducibility: Sufficient details are provided. the authors do not mention if they will release the code of their attacks.

**Strength And Weaknesses:**

Strengths:

1. The threat model considered here is novel and interesting and may lead to more work in this area.

2. The paper is well-organized, easy to read, and accessible to a broad audience.

3. There is enough technical novelty in the design of probabilistic attacks and the formulation of mini-max defense for multivariate forecasting models for publication at ICLR.

4. The experimental evaluation supports the main claims made in the paper. I provide some suggestions below to make the evaluation more compelling.

Weaknesses:

1. The attacks assume an offline setting. It is not clear if the attacks work in an online setting where the time series data is generated on the fly. This is particularly relevant for stock market predictions.


**Summary Of The Paper:**

This paper studies adversarial attacks on multivariate forecasting models - a problem that has not been explored in prior work but is practically relevant due to the prevalence of these models in various real-world tasks. The authors first propose sparse, white-box deterministic indirect attacks that add adversarial perturbations to a time series different from the target series.  To make the attack hard to detect, probabilistic attacks. Next, the authors design two defenses in this setting based on randomized smoothing and mini-max optimization. Experimental evaluation is performed on a variety of datasets. The results show that probabilistic attacks are more effective than deterministic ones while the defenses enable more robustness against these attacks than no defense.

**Summary Of The Review:**

In summary, this is a well-written paper that tackles an important but unexplored problem in the literature that should lead to more work in this area. The technical claims are novel and the contributions are supported by evaluation results. I believe that this paper is a good candidate for acceptance at ICLR.

---

> ### Author Response · Authors · 2022-11-15
> **Response to Reviewer qQby**
>
> Thanks for your comments and strong support for our work. We would like to address your concerns with further clarification below with direct reflection on the revision.
>
> # 1 Attack other time series
> First, we did study the effect of different hyper-parameters, like varying target items, attacked time horizon, sparsity level, and noise level. We also experimented on various benchmark time series datasets (including electricity, traffic, taxi, wiki). To be specific on varying target items,
> we reported the attack & defense effect when multiple time series can be attacked, see our Table 2, where we select **TS1 and TS5** to attack simultaneously. Under these settings, we can still arrive at the same conclusion:
> * Our attacking scheme is effective as the attack can increase wQL by up to 103%.
> * Our defense mechanisms (randomized smoothing & mini-max) outperform baseline (no defense & data augmentation).
>
> # 2 Perturbation threshold
> We did report the perturbation threshold in Page 13 of appendix. To be specific, we follow the convention of previous work to set the perturbation threshold $\delta=c_1|x|$ and choose $c_1=0.5$.
>
> # 3 Randomized smoothing certification
> Yes, our randomized smoothing is a certified defense mechanism as shown in Theorem 4.1 on Page 6. This theorem states that after randomized smoothing, the resulting output distribution of DeepVAR will be a Lipschitz continuous function measured by function $L_\infty$-norm. This Lipschitz constant is also given in the theorem, which is $\sqrt d/\sigma$.
>
> # 4 Runtime
> The runtime for regular DeepVAR training is around 10 minutes.
> The runtime for our deterministic attack is around 3 minutes.
> The runtime for probabilistic attack is around 3 minutes.
> The runtime for mini-max defense training of DeepVAR is around 20 minutes.
>
> To summarize, mini-max defense takes longer training time but is more capable of defending both deterministic and probabilistic attack (see Table 1,2,7,8,9). Note that RS is a post-processing technique and doesn’t require to re-train the model. However, it does increase the inference time due to more required sample paths.
>
> # 5 Online setting
> We thank the reviewers for proposing another interesting setting. Albeit we only consider off-line setting in this paper, here we describe some basic idea as an extension:  With data generated on the fly, we can separate a validation set containing the most recent data for hyper-parameter tuning for defense mechanisms (and attacks therein). In this way, the tuning parameters can be updated in real time within the latency limit.
>
> We believe we have addressed all your concerns. Please don’t hesitate to let us know if you have any additional comments or questions in the rebuttal period.

---

### Author Response · Authors · 2022-11-15
**General Response**

We sincerely thank the reviewers for valuable discussions to help us improve our manuscripts. Specifically, we appreciate the reviewers giving credit on our suggested new and important topic in time series setting (by Reviewer qQby, Reviewer oCqG, Reviewer K2Yj); recognizing our technical novelty and contribution of developing probabilistic attacks and mini-max defense (by Reviewer qQby and Reviewer K2Yj); pointing out extensive and solid the experiments that support our claim (by Reviewer qQby, Reviewer hzNw and Reviewer oCqG). Next, we would like to clarify the following concerns.

# Novelty
Even with these credits commonly mentioned across reviewers, we would like to reiterate the highlight of our novelty in three-fold: (i). We propose a novel and interesting question, **sparse indirect** attack, which is to harm the prediction of one TS by attacking other TS. This question is only applicable under multivariate time series setting, not in image classification or univariate time series as discussed by prior work since there is no correlation in these settings (summarized in the last paragraph of Page 1). (ii). We propose deterministic attack and probabilistic attack to achieve this goal. Although the way of sparsifying deterministic attack may be similar to that used in the classification setting (already credited in Sec 3.2 of our revision), the objective function is designed differently for the distinct task of sparse attack: in classification, the attack is designed to make the classifier give a wrong prediction; but in our work, the attack aim to mount an **indirect** attack on the other TS. Besides, we also propose a novel differentiable probabilistic attack to achieve sparse indirect attack using a sparse layer, which has not been investigated in this literature. (iii). We develop two defense mechanisms , randomized smoothing (RS) and mini-max defense. Although RS is generalized from prior work ([Yoon et al, 2022]), we use different and novel techniques to prove a more general robustness guarantee which holds for any $\delta$ (vs previous work [Yoon et al, 2022] whose certification only holds for $\delta\to0$. Kindly see the Remark at the top of Page 7. ). Moreover, we propose a mini-max defense that beats baseline defense mechanisms (data augmentation) and even outperforms our own randomized smoothing in most of the cases.

# Reproducibility
About the source code, we will release the cleaned one after revision through the github repository for reproducibility.

---

### Decision · Program_Chairs · 2023-01-20

**Decision:**

Accept: poster

**Justification For Why Not Higher Score:**

The work is good, but not good enough to deserve spotlight acceptance due to our mixed feelings regarding its technical novelty.


**Justification For Why Not Lower Score:**

After the discussion period, all four reviewers are supportive of acceptance (6, 6, 6, 8) with sound justification. I can hardly go against their recommendation to reject the paper.


**Metareview: Summary, Strengths And Weaknesses:**

Although we have mixed feelings regarding the novelty of this work with respect to the problem formulation and the actual design of the attacks and defense formulation, we do recognize the novelty of the threat model which has potential to arouse further research interests in this area. The main claims in the paper are also well supported. Nevertheless, the paper has room for improvement before final publication. We recommend the authors to consider our comments and suggestions thoroughly when revising their paper. Some potential longer-term extensions, such as generalizing from the offline to online setting, may also be discussed.


**Note From Pc:**

if the above contains the word "oral" or "spotlight" please see: "oral" presentation means -> notable-top-5% and "spotlight" means -> notable-top-25%. As stated in our emails, we are disassociating presentation type from AC recommendations